# Prophage-like gene transfer agents promote *Caulobacter crescentus* survival and DNA repair during stationary phase

**Kevin Gozzi[1]¤a, Ngat T. Tran[2], Joshua W. Modell[1]¤b, Tung B. K. Le[2]\*, Michael T. Laub[1,3]\***

**1** Department of Biology, Massachusetts Institute of Technology, Cambridge, Massachusetts, United States of America, **2** Department of Molecular Microbiology, John Innes Centre, Norwich, United Kingdom, **3** Howard Hughes Medical Institute, Massachusetts Institute of Technology, Cambridge, Massachusetts, United States of America

¤a Current address: Rowland Institute at Harvard, Harvard University, Cambridge, Massachusetts, United States of America
¤b Current address: Department of Molecular Biology & Genetics, Johns Hopkins University School of Medicine, Baltimore, Maryland, United States of America
* tung.le@jic.ac.uk (TBKL); laub@mit.edu (MTL)

**Data Availability Statement:** All relevant data are within the paper and its Supporting information files, except for RNA-seq, PacBio sequencing, and ChIP-seq data which are available at GEO with the

## Abstract

Gene transfer agents (GTAs) are prophage-like entities found in many bacterial genomes that cannot propagate themselves and instead package approximately 5 to 15 kbp fragments of the host genome that can then be transferred to related recipient cells. Although suggested to facilitate horizontal gene transfer (HGT) in the wild, no clear physiological role for GTAs has been elucidated. Here, we demonstrate that the α-proteobacterium *Caulobacter crescentus* produces bona fide GTAs. The production of *Caulobacter* GTAs is tightly regulated by a newly identified transcription factor, RogA, that represses *gafYZ*, the direct activators of GTA synthesis. Cells lacking *rogA* or expressing *gafYZ* produce GTAs harboring approximately 8.3 kbp fragment of the genome that can, after cell lysis, be transferred into recipient cells. Notably, we find that GTAs promote the survival of *Caulobacter* in stationary phase and following DNA damage by providing recipient cells a template for homologous recombination-based repair. This function may be broadly conserved in other GTA-producing organisms and explain the prevalence of this unusual HGT mechanism.

## Introduction

Horizontal gene transfer (HGT) is a powerful and common process in bacteria, enabling the facile acquisition of new DNA that can rapidly and dramatically alter the physiology or survival of a cell. HGT impacts both the short-term adaptability of bacteria and their long-term patterns of evolution [1,2]. Bacterial HGT mechanisms traditionally fall into 3 categories: transformation or natural competence, bacteriophage-mediated transduction, and direct cell–cell conjugation. A fourth, less studied mechanism involves gene transfer agents (GTAs), which were first identified and characterized in the α-proteobacterium *Rhodobacter capsulatus* [3]. GTAs are thought to derive from defective, "domesticated" prophages that mediate gene transfer via phage-like protein capsids filled with double-stranded DNA [4].

respective accession numbers GSE184480, GSE184478, and GSE184477.

**Funding:** This work was supported by a National Science Foundation Graduate Research Fellowship to K.G., the Royal Society University Research Fellowship (URF\R\201020) to T.B.K.L., the BBSRC grant-in-add (BBS/E/J/000PR9791 to the John Innes Centre) to N.T.T., and an NIH grant to M.T.L. (R01GM082899), who is also an Investigator of the Howard Hughes Medical Institute. The funders had no role in study design, data collection and analysis, decision to publish, or preparation of the manuscript.

**Competing interests:** The authors have declared that no competing interests exist.

**Abbreviations:** CFU, colony-forming unit; ChIP-seq, chromatin immunoprecipitation paired with deep sequencing; EMS, ethyl methanesulfonate; GTA, gene transfer agent; HGT, horizontal gene transfer; HU, hydroxyurea; MMC, mitomycin C; PVDF, polyvinylidene fluoride; SPR, surface plasmon resonance; TEM, transmission electron microscopy.

In contrast to phages, which usually preferentially package their own DNA, GTA particles package genomic DNA fragments relatively nonspecifically [5,6]. In some cases, the packaged DNA may be depleted for the GTA-encoding locus due to high levels of local transcription that occlude the packaging machinery [5]. Notably, the length of the DNA fragment packaged into a GTA is not sufficient to contain the entire GTA locus, thus GTAs cannot transfer themselves horizontally [7]. Moreover, in *R. capsulatus* and likely other GTA-producing species, several genes necessary for GTA production are separated in the genome from the locus encoding the structural components of the GTA [8]. In short, GTAs are not infectious phage-like particles and instead are likely a means of disseminating DNA to other cells in a population. However, their properties and functions remain very poorly understood.

Genome sequence analysis suggests that GTAs are widespread in the α-proteobacteria and were present in one of the common ancestors of this clade [9]. Given the numerical dominance of α-proteobacteria in marine environments, GTAs may be critical in such environments where HGT is extremely prevalent [10]. GTAs have been discovered in some other taxa, including the δ-proteobacterium *Desulfovibrio desulfuricans*, the spirochete *Brachyspira hyodysenteriae*, and the archaeal methanogen *Methanococcus voltae*, but they have not been as closely examined in those organisms [11–14].

The physiological benefit(s) of GTAs remain unclear. For *R. capsulatus*, GTAs are produced during starvation or other stress conditions, whereas for *Bartonella grahamii*, the fastest growing cells in a population are both the producers and receivers of GTAs [6,15,16]. However, in neither case have GTA-producing cells been shown to harbor an advantage over non-producers. GTAs are frequently postulated to spread beneficial alleles through a population [6,7], but evidence supporting this function is lacking. Moreover, mathematical modeling indicates that it is more favorable for a population if cells that acquire a beneficial allele simply grow and propagate the allele via vertical inheritance rather than distribute the allele via GTAs [17]. This work further suggested that in a mixed population of GTA producer and non-producers, the non-producers will eventually take over.

Here, we identified and characterized the GTA produced by the α-proteobacterium *Caulobacter crescentus*, an organism traditionally studied to understand cell cycle regulation and cellular asymmetry in bacteria. We found that GTA production in *Caulobacter* is tightly regulated by the transcription factor RogA, which directly represses 2 genes, *gafYZ*, that are necessary and sufficient, if expressed together, to activate GTA transcription. Cells lacking *rogA* or overexpressing *gafYZ* transcribe the primary, 21-gene GTA locus and several auxiliary loci, which combine to produce a particle that packages genomic DNA fragments of approximately 8.3 kbp. The production of GTAs leads to cell lysis and the release of particles into the surrounding environment. These released GTAs can transfer genetic material into a recipient cell, which can recombine the GTA-delivered DNA onto its chromosome. Importantly, we find that GTA production allows *C. crescentus* to better survive in stationary phase and to tolerate DNA damage during stationary phase by using GTA-derived DNA as a template for homologous recombination-based repair. Collectively, our work provides the first experimental characterization of *Caulobacter* GTAs and now demonstrates a physiological benefit of a GTA, which may be shared by other GTA-producing organisms.

## Results

### Identification of RogA, a LexA-like repressor, in *Caulobacter crescentus*

We previously identified a RecA-independent mechanism by which *C. crescentus* cells respond to DNA damage, particularly double-strand breaks, culminating in activation of the transcription factor DriD in an SOS-independent manner [18] (Fig 1A). To identify additional factors

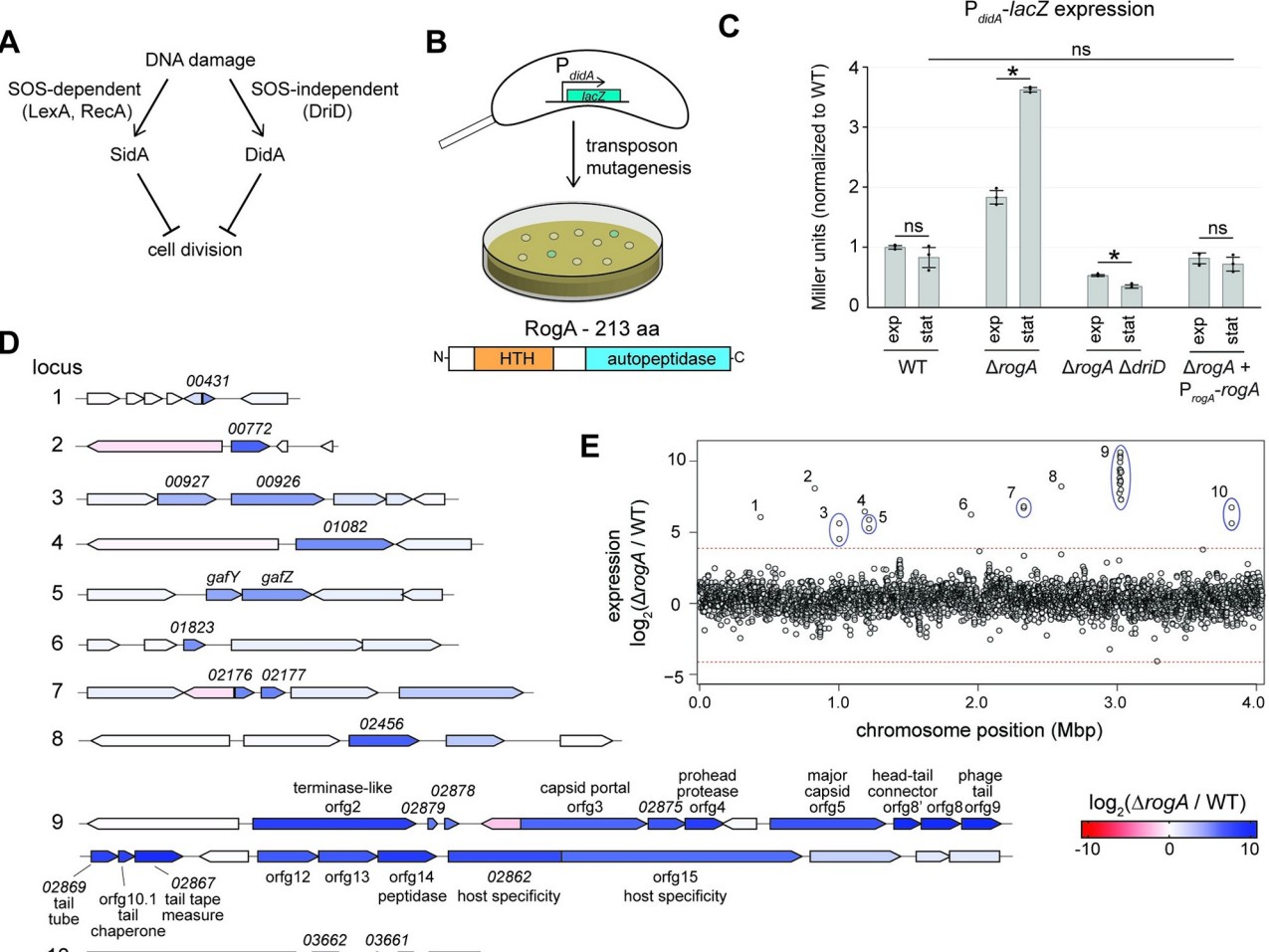

**Fig 1. Identification of a GTA repressor.** (A) Schematic of the 2 DNA damage response pathways in *Caulobacter crescentus*. (B) Diagram of a transposon mutagenesis screen to identify mutants with altered expression of *didA*. Schematic of predicted RogA domains based on homology to known protein domains. (C) β-galactosidase assay measuring transcriptional activity of a P$_{didA}$-*lacZ* reporter with different genetic backgrounds ($n = 3$, error bars indicate SD). Cells were assayed in both exponential and stationary phase. * = $p$-value <0.05 between indicated data. ns = not significant. Data are available in S1 Data. (D) Transcriptomic analysis comparing wild-type to Δ*rogA* cells in stationary phase using RNA-seq. Genetic loci with a log$_2$ fold-change greater than 4 are shown with each gene's expression color-coded (see legend). (E) The log$_2$ fold-change of each gene's expression in the Δ*rogA* mutant relative to the wild type is plotted, with each gene's genome position on the x-axis. Genetic loci with a log$_2$ fold-change greater than 4 are shown with each gene's expression color coded. Dotted line is equal to a log$_2$ fold-change of 4. Mean log$_2$ fold-change of all genes was 0.36 ± 1.01. Data are available in S1 Data. GTA, gene transfer agent.

involved in this pathway, we performed transposon mutagenesis on a strain harboring a reporter of DriD activity in which the promoter of the target gene *didA* was fused to *lacZ* (Fig 1B). In the absence of DNA damage, wild-type colonies harboring the reporter appear white, indicating low basal *didA* promoter activity. A loss-of-function mutation in a negative regulator was predicted to result in high promoter activity and blue colonies. After screening >10,000 colonies, multiple independent insertions were identified in *CCNA_00080*, a previously unannotated gene predicted to encode a 213-amino acid protein. Bioinformatic analyses suggested that *CCNA_00080,* now named RogA (Repressor of GTA activators), encodes a putative LexA-like transcriptional repressor with an N-terminal helix-turn-helix DNA-binding domain and a C-terminal LexA/cI-like autopeptidase domain though key residues required for cleavage of LexA/cI are missing (Fig 1B).

To verify that RogA negatively regulates *didA* expression, we generated a marked *rogA* deletion strain (Δ*rogA*::*tet*[R]) in which all but the first 5 and last 5 codons of *rogA* were replaced by the tetracycline resistance cassette. Using our P*didA*-*lacZ* reporter, we found that this Δ*rogA* mutant had approximately 4-fold higher *didA* promoter activity in stationary phase with a more modest 2-fold increase in exponential phase (Fig 1C). These increases in *didA* expression were complemented by providing a wild-type copy of *rogA* on a low-copy plasmid. The elevated expression of *didA* in the Δ*rogA* mutant was abolished when *driD* was deleted (Fig 1C). Compared to wild-type, further DNA damage in the Δ*rogA* mutant during early stationary phase did not result in an increase of *didA* expression, suggesting *rogA* is upstream of DNA damage activation of this pathway (S1 Fig). Together, these findings suggest that RogA indirectly regulates *didA* by somehow modulating DriD's expression or activity, possibly by causing an accumulation of double-stranded DNA breaks and ssDNA, which are known to activate DriD [18,19].

## RogA controls a GTA by repressing its activators GafY and GafZ

To identify other genes regulated by RogA, we used RNA-seq to examine transcriptional changes in a Δ*rogA* mutant compared to the wild-type strain. Notably, we observed strong (approximately 500 to 1,000 fold) up-regulation of a 21-gene cluster (locus 9 in Fig 1D and 1E) predicted to encode a putative GTA similar to that found in *R. capsulatus* (S2A Fig). The GTA of *C. crescentus* has never been characterized and bioinformatic studies have suggested that *C. crescentus* may have an incomplete GTA cluster [20]. In addition to the GTA cluster, we also observed approximately 40-fold up-regulation of an operon consisting of *CCNA_01111* and *CCNA_01112* (locus 5 in Figs 1D and 1E and S2B), which are homologous to the N- and C-terminal portions, respectively, of the GTA activator *gafA* in *R. capsulatus* [21]. *CCNA_01111* was previously named *hdaB* as overexpressing this gene can impact chromosomal DNA replication [22]; however, given the homology to GafA and the results presented herein, we propose GafY and GafZ as names for *CCNA_01111* and *CCNA_01112* for consistency with *R. capsulatus*.

To better understand the prevalence of GafY, GafZ, and RogA, we searched for homologs across 1,331 α-proteobacteria genomes. Homologs of RogA have not previously been implicated in GTA regulation in other species. However, RogA is highly conserved in α-proteobacteria, with at least 1 homolog in 819 of the 1,331 sequenced genomes. Additionally, RogA commonly co-occurs with core GTA genes (*CCNA_02880–02861*) and auxiliary gene homologs such as *gafYZ* and *CCNA_02456* (S2C Fig). We identified 759 GafZ homologs with 494 containing an adjacent GafY homolog. GafY and GafZ homologs were also found more often as separate genes than fused into one, as in *R. capsulatus*.

As noted above, DriD was identified as an activator of *didA* in response to double-stranded DNA breaks [18]. Given that maturation of bona fide GTA particles would involve the production of linear DNA fragments, we hypothesized that such fragments drive DriD-dependent activation of *didA* in the Δ*rogA* mutant (Fig 1C). To test if GTA-packaged DNA was accumulating in the Δ*rogA* mutant, total DNA was isolated from cells in stationary phase, which was when *didA* expression was highest in the Δ*rogA* mutant (Fig 1C), and examined by gel electrophoresis (Fig 2A). In addition to the bright, low-mobility band representing genomic DNA, we observed a second band approximately 8 to 10 kb in length for DNA extracted from the Δ*rogA* mutant, but not wild-type cells or Δ*rogA* cells complemented with wild-type *rogA* (Fig 2A). This putative GTA DNA in a Δ*rogA* mutant was not seen if the primary GTA locus was also deleted (Fig 2A). Similarly, the putative GTA band was not seen in a Δ*rogA* mutant also harboring a deletion of either of the putative GTA activators *gafY* and *gafZ* (Fig 2A).

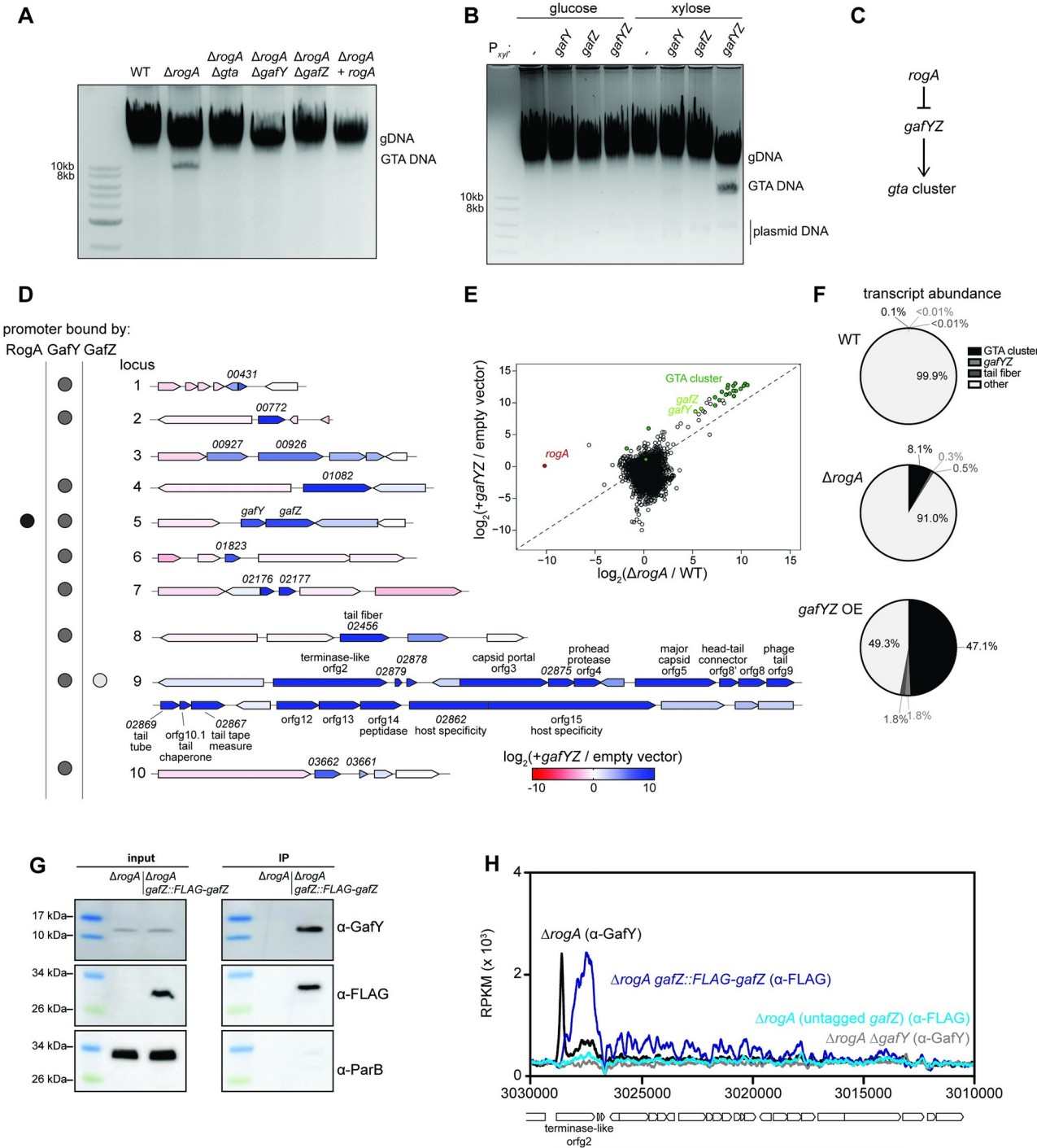

**Fig 2. Characterization of the GTA regulatory pathway.** (A) Total DNA extraction from different strains grown up to stationary phase. DNA was purified and separated via gel electrophoresis on a 1% agarose gel. (B) Same as A with total DNA extracted from strains bearing a high-copy plasmid with the xylose promoter driving expression of either nothing (empty), *gafY* alone, *gafZ* alone, or both *gafY* and *gafZ* in their native arrangement. (C) Simple genetic diagram of how the *gta* gene cluster is hypothesized to be regulated. (D) Transcriptomic analysis comparing *gafYZ* overexpression to an empty vector control in stationary phase using RNA-seq. The 10 up-regulated loci from the Δ*rogA* RNA-seq are shown with their corresponding shading for the log2 fold change of each gene during *gafYZ* overexpression. Binding of RogA, GafY, and GafZ, as determined by ChIP-seq, are shown for each locus' promoter by a black, gray, or white dot, respectively. (E) Log2 fold change of each gene during *gafYZ* overexpression compared to an empty vector is plotted as a function of the gene's log2 fold change in a Δ*rogA* mutant compared to the wild-type strain. Genes in the GTA cluster are highlighted in dark green, *gafY* and *gafZ* in light green, and *rogA* in red. $R^2$ of the GTA cluster genes and *gafYZ* (log2 transformed) is 0.92. Data are

available in S1 Data. (F) Transcript abundance of 4 different categories out of all transcripts in different RNA-seq samples. Wedges are colored according to the legend. Data are available in S1 Data. (G) Western blot analysis of co-immunoprecipitation of GafY with immunoprecipitation of Flag-GafZ. ParB served as a non-associated protein control. (H) ChIP-seq profiles of GafY with an anti-GafY antibody in either Δ*rogA* (black) or Δ*rogA* Δ*gafY* (gray) and FLAG-GafZ with an anti-FLAG antibody in either Δ*rogA gafZ::FLAG-gafZ* (dark blue) or Δ*rogA* (light blue). Profiles were plotted with the x-axis representing genomic positions and the y-axis representing the number of reads per kilobase pair per million mapped reads (RPKM) using custom R scripts. Data are available in S1 Data. GTA, gene transfer agent.

Given that *gafY* and *gafZ* were necessary for GTA DNA band production in a Δ*rogA* mutant, we tested whether these 2 genes were also sufficient to induce GTA production. The *gafY* and *gafZ* coding regions were cloned either individually or together on a high-copy plasmid, with expression driven by a xylose-inducible promoter. The GTA band strongly accumulated only when both *gafY* and *gafZ* were induced (Fig 2B). In this experiment, *rogA* was still present, indicating that *gafY* and *gafZ* act downstream of *rogA*, with RogA likely serving to repress *gafYZ* expression and GafYZ acting as an activator of the GTA gene cluster (Fig 2C). Overexpression of *gafYZ*, and *gafY* alone to a lesser degree, in stationary phase was also sufficient to activate expression of the *didA* promoter, further suggesting that GTA production leads to DNA strand breaks throughout the cell (S2D Fig).

To better understand the role of GafYZ in activating the *Caulobacter* GTA system, we performed RNA-seq on cells harboring a xylose-inducible copy of *gafYZ* or an empty vector, each treated with xylose for 3 h in stationary phase. Cells overexpressing *gafYZ* exhibited a >1,000-fold induction of genes in the GTA cluster (Fig 2D). We also observed strong induction of several loci distal to the GTA-encoding cluster (Fig 2D, loci #1–8, 10). We observed robust up-regulation of *ccna_02456*, which has homology to the GTA tail fiber gene in *R. capsulatus*. Locus 4 contained the gene *ccna_01082*, which is predicted to encode a homeodomain-related protein with no known function. The remaining up-regulated genes in loci #1–3, 5–8, and 10 contained hypothetical genes with no known function.

Consistent with the notion that GafY and GafZ function downstream of RogA, we observed a high correlation between the genes most up-regulated in cells overexpressing *gafYZ* or lacking *rogA* ($r^2$ = 0.92; Fig 2E). In both cases, GTA activation led to a dramatic shift in bulk gene expression, with more than 8% of all mRNA reads for the Δ*rogA* mutant mapping to the GTA cluster and accessory genes, and approximately 49% for *gafYZ* overexpression, compared to <0.1% in wild-type cells (Fig 2F). Collectively, these results indicate that RogA represses GTA expression and that *gafYZ* are necessary and sufficient to induce GTA expression.

## RogA directly regulates *gafYZ* and GafYZ directly regulate GTA transcription

In *R. capsulatus*, the GTA activator GafA is a single polypeptide [21], whereas in *C. crescentus*, the GafA homolog is split into 2 proteins, GafY and GafZ (S2B Fig). To test whether GafY and GafZ form a complex, we performed a co-immunoprecipitation (IP) with a Δ*rogA* strain expressing an N-terminal FLAG-tagged version of *gafZ* from its native promoter. Anti-FLAG antibodies were used to IP FLAG-GafZ from cell lysate and the resulting eluate was then blotted with an anti-GafY polyclonal antibody. We observed significant enrichment of GafY in the IP compared to the pre-IP input control (Fig 2G). Another cytoplasmic protein, ParB, was not enriched in the IP sample, suggesting that the enrichment of GafY was specific.

We hypothesized that RogA and GafY, which encode DNA-binding proteins, and potentially GafZ, act directly as transcription factors at target promoters. To determine their direct targets, we used chromatin immunoprecipitation paired with deep sequencing (ChIP-seq). For RogA and GafY, we used polyclonal antibodies raised against each protein and compared wild-type cells to the corresponding deletion strain. For GafZ, we used an anti-FLAG antibody

and compared cells producing FLAG-GafZ to isogenic cells producing untagged GafZ. GafY was enriched immediately upstream of all but 1 of the 10 loci that were most up-regulated during GTA expression (Δ*rogA* and *gafYZ* overexpression), including the major GTA cluster and *gafYZ* itself (Fig 2D and 2H and S3A Fig). GafZ was found only within the promoter and coding regions of the major GTA cluster, with the highest occupancy within *CCNA_02880*, the first gene of the GTA cluster (Fig 2D and 2H and S3B Fig). RogA was enriched only at the promoters of *gafYZ* and *CCNA_02002*, a gene whose expression did not change during GTA induction (Fig 2D and S3C Fig).

To validate our ChIP-seq analysis, we tested whether purified RogA (see Methods) can bind the *gafYZ* promoter in vitro. Using overlapping approximately 40-bp DNA fragments, we scanned the *gafYZ* promoter for RogA binding via surface plasmon resonance (SPR). We observed strong binding of RogA to 3 DNA fragments that span the predicted core promoter region for *gafYZ*, consistent with RogA being a repressor of this locus (S4A and S4B Fig). This region featured 2 inverted repeats of GGAA-N$_4$-TTCC (S4C Fig), as did the promoter of *CCNA_02002*. We also observed strong interaction between purified GafYZ complex and the GTA promoter region (S4D Fig). We were unable to predict a binding motif for GafYZ in the promoters of target genes.

Taken all together, our results demonstrate that RogA silences GTA production by binding directly to and repressing the promoter of *gafYZ*, which encode the direct activators of the GTA cluster and several auxiliary genes and operons.

## The *Caulobacter* GTA encapsulates random genomic DNA and triggers cell lysis

To determine whether *C. crescentus* cells can produce functional GTA particles, we first sought to characterize the DNA fragments produced in cells lacking *rogA*. We isolated approximately 8 to 10-kbp DNA fragment produced by Δ*rogA* cells (Fig 2A) and then digested this DNA with SalI, a restriction endonuclease with thousands of restriction sites across the *C. crescentus* genome. The SalI-digested DNA ran as a smear with no distinct bands, suggesting that the GTA DNA is heterogeneous, as a less complex input would generate distinct bands (S5A Fig). We also used qPCR to show that various genomic regions could be amplified using the putative GTA DNA as template (S5B Fig), further indicating that the GTA DNA is heterogeneous.

To more precisely assess the GTA DNA, we isolated it from *gafYZ*-overexpressing cells in stationary phase and then sequenced it using PacBio Long-Read DNA sequencing. The GTA DNA fragments were 8.3 kb on average and always comprised a single, continuous region of the *C. crescentus* genome. Notably, we detected all regions of the genome, though with approximately 10-fold range in relative abundance (Fig 3A). Two broad peaks were centered around 1.0 and 3.0 Mbp, the midpoints of the 2 arms of the chromosome, with sequencing depth minima near *oriC* and *ter*. There was no major decrease in packaging of the GTA locus itself (Fig 3A inset).

GTA production in *R. capsulatus* causes lysis and death of the producing cells [23]. Similarly, we found that inducing *gafYZ* in *C. crescentus* led to approximately 100-fold decrease in viable cells after 60 min and approximately 1,000-fold decrease after 4 h (Fig 3B) with a concomitant increase in extracellular protein throughout the time course, indicative of cell lysis (Fig 3C). The lethality of overexpressing *gafYZ* was completely GTA-dependent (Fig 3B). To ensure that the lethality observed was not an artifact of overexpressing *gafYZ*, we also tracked cell lysis as Δ*rogA* and wild-type cells entered stationary phase. We observed a marked increase in cell lysis for the Δ*rogA* strain, as measured by protein release into the supernatant, as cells reached an OD$_{600}$ of approximately 1.3 (Fig 3D), matching the stationary phase-dependence

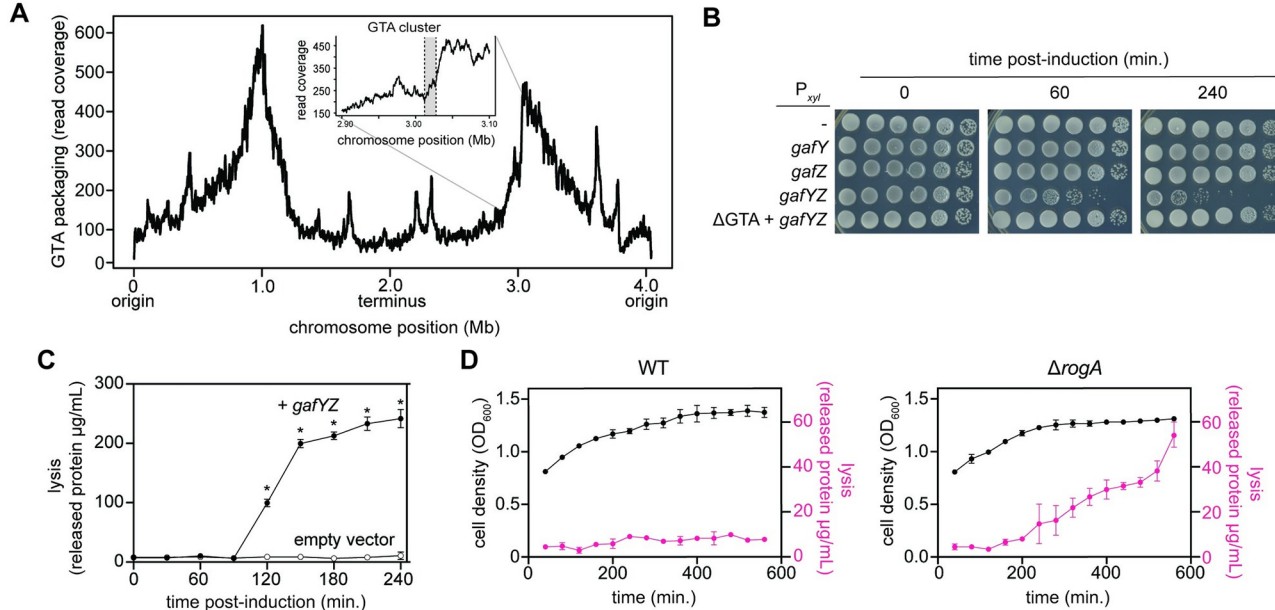

**Fig 3. Induction of the GTA cluster leads to GTA packaging and lysis.** (A) Long-read (PacBio) sequencing of DNA packaged during GTA production. Inset displays the packaging of the GTA cluster, in gray, and the surrounding DNA. Data are available in S1 Data. (B) Survival of cells bearing high-copy plasmids driving xylose-inducible expression of different genes. Cells were induced with 0.3% xylose and CFUs were enumerated over time. (C) Quantification of released protein content in supernatant during induction of *gafYZ* over time in strains bearing either $P_{xyl}$-*gafYZ* (black circles) or empty vector (white circles) with 0.3% xylose in PYE ($n = 3$, error bars indicate SD). * = *p*-value <0.05 between data of same time point. Data are available in S1 Data. (D) Quantification of released protein content in supernatant (magenta) as wild-type and Δ*rogA* cells enter stationary phase ($n = 3$, error bars indicate SD). Optical density of cells is plotted in black. Data are available in S1 Data. CFU, colony-forming unit;, GTA, gene transfer agent.

of our *didA* reporter (Fig 1C) in the Δ*rogA* mutant. We also observed a significant decrease in colony-forming units (CFUs) in the Δ*rogA* mutant compared to wild-type cells as they entered stationary phase, further suggesting GTA production correlates with cell lysis and lethality (S5C Fig). To test that cells were indeed lysing, we stained *gafYZ* overexpressing cells with propidium iodide to track cell death as GTA production was induced. We observed a similar appearance of dead cells after 2 h of induction (S5D Fig), in line with the increase in protein release observed (Fig 3C). No homologs of lysis-associated genes in *R. capsulatus* [5] were identified in the *C. crescentus* genome nor do any of the genes up-regulated during GTA expression have homology to holin/endolysin genes.

## GTAs can transfer DNA onto the chromosome of recipient *C. crescentus* cells

To determine if the cell lysis observed was associated with the release of GTA particles, we added SYBR-gold, a DNA-staining dye, to agarose pads and imaged cells harboring an inducible copy of *gafYZ*. In cells expressing *gafYZ*, there were many SYBR-gold positive, phase-contrasting particles smaller than an individual cell (Fig 4A). These particles could be precipitated with polyethylene glycol and separated from other cellular components after centrifugation through a sucrose gradient (S5E Fig) with the putative capsid proteins producing a banding pattern by SDS-PAGE reminiscent of that seen with *R. capsulatus* GTAs [24] (S5F Fig). Transmission electron microscopy (TEM) revealed many spherical particles in the supernatant of Δ*rogA* cells, but not wild-type cells (Fig 4B). These particles were approximately 40 nm in diameter with a spherical head and no visible tail, despite the presence of genes predicted to

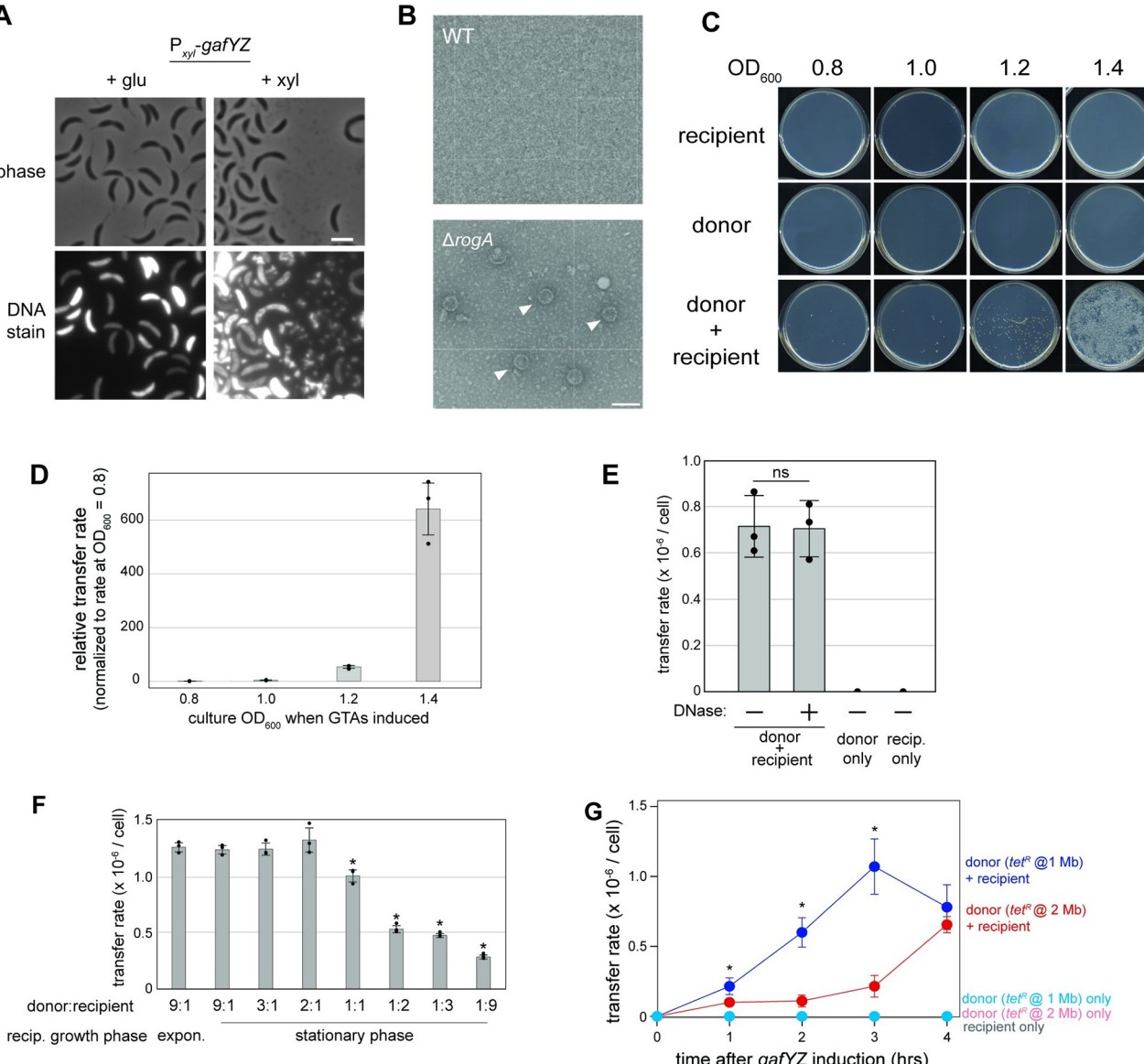

**Fig 4. GTAs are released into the supernatant and can transfer genetic material.** (A) Micrographs of cells bearing P$_{xyl}$-*gafYZ* on a high-copy vector grown with 0.2% glucose or 0.3% xylose for 3 h. Samples were stained with SYBR-gold and imaged for fluorescence. Scale bar = 2 μm. (B) Transmission electron micrographs of filtered supernatant from wild-type or Δ*rogA* cells in stationary phase. Scale bar = 100 nm. (C) Representative images of gene transfer with induction of *gafYZ* at different optical densities. (D) Quantification of the gene transfer rate of the *tet*$^R$ resistance marker from a donor strain bearing P$_{xyl}$-*gafYZ* on a high-copy plasmid (*n* = 3, error bars indicate SD). Transfers were carried out with induction of *gafYZ* at different culture densities as marked. Transfer rates are shown relative to transfer measured at cell density = 0.8. Data are available in S1 Data. (E) GTA-mediated gene transfer rates with or without addition of DNase. A total of 5 U/mL DNase I was added to donor culture for 1 h following induction, followed by 3 h co-culture with recipient strain (*n* = 3, error bars indicate SD). ns = not statistically significant via *t* test. Data are available in S1 Data. (F) GTA-mediated gene transfer rates with donor and recipient cells at different ratios (*n* = 3, error bars indicate SD). Recipient cells were also assayed when in exponential phase. * = *p*-value <0.05 comparing data to "9:1 stationary" sample. Data are available in S1 Data. (G) Transfer rate of the *tet*$^R$ resistance marker from donors bearing P$_{xyl}$-*gafYZ* on a high-copy plasmid and a *tet*$^R$ resistance marker either at a region of high packaging (1.0 Mb) or low packaging (2.0 Mb) (*n* = 3, error bars indicate SD). Each donor and recipient strain were grown either alone or together in the presence of 0.3% xylose. * = *p*-value <0.05 comparing data of same time point between the "donor (*tet*$^R$ @ 1 Mb) + recipient" and "donor (*tet*$^R$ @ 2 Mb) + recipient" samples. Data are available in S1 Data. GTA, gene transfer agent.

encode a tail tube, a hub/baseplate protein, and tail fibers (Fig 1D and S2A Fig). It is possible the tail is too short to resolve, which would differ from the long tails of *RcGTA* [25], or that sheer forces during the preparation for TEM led to loss of tails, in line with the observed sensitivity of GTAs to various purification methods [3]. Similar particles were also observed in PEG-precipitated supernatant from cells overexpressing *gafYZ* (S5G Fig).

To test whether these particles could mediate genetic transfer, we generated "donor" strains harboring an inducible copy of *gafYZ* to drive GTA expression and a tetracycline resistance marker at chromosome position 1.0 Mb. The donor strain was then mixed with a "recipient" strain bearing a chloramphenicol resistance marker and inducer (0.3% xylose) was added. The co-incubated cells were plated after 4 h, selecting for doubly resistant colonies indicative of successful transfer. For donor or recipient cells incubated alone, no colonies were detected (Fig 4C). In contrast, when mixing donor and recipient cells, doubly resistant colonies emerged, with the highest number arising when GTA production was induced in stationary phase at $OD_{600}$ = 1.4 (Fig 4C and 4D). The maximum rate of transfer of the $tet^R$ cassette from the highly packaged 1.0 Mb position in these conditions was approximately $1 \times 10^{-6}$ per recipient cell. A key feature of GTA-mediated transfer is that transfer is DNase resistant [26]. Indeed, we observed no difference in transfer rates when the supernatant was treated with excess DNase to digest any exogenous DNA during co-incubation (compared to no DNase added), indicating that transfer does not occur through transformation by free DNA (Fig 4E).

To test whether transfer required recipient cells to also be in stationary phase, we mixed stationary phase donor cells with either exponential or stationary phase recipient cells. There was no major difference in the transfer rate of $tet^R$ (Fig 4F). We also varied the donor:recipient cell ratio and found that the rate of transfer decreased once recipients outnumbered donors (Fig 4F). Finally, we tested transfer from a donor harboring a $tet^R$ cassette at 2.0 Mb, a genomic region that was packaged less frequently based on our long-read sequencing (Fig 3A). The efficiency of transfer was approximately 5- to 10-fold lower from this strain at 2 and 3 h post-induction of *gafYZ*, though comparable after 4 h (Fig 4F). Taken all together, our results demonstrate that *Caulobacter* cells can produce functional GTA particles capable of transferring DNA from one cell to another. Donor cells must be in stationary phase to produce functional GTA particles but can transfer to cells in either exponential or stationary phase.

## GTAs promote survival in starvation conditions

We next sought to determine if there was any physiological benefit to GTAs in *C. crescentus*. Because GTA production and transfer increased upon entry to stationary phase, we compared wild-type and Δ*rogA* strains during stationary phase. For longer-term co-incubations, the Δ*rogA* mutant was used as opposed to *gafYZ* overexpression, as overexpression over long time periods would require inducer to be added several times because the inducer (xylose versus glucose) gets consumed. Over the course of 120 h, we observed approximately 10- to 30-fold increase in the survival of the GTA-producing Δ*rogA* mutant compared to wild-type cells (Fig 5A and 5B). Expressing *rogA* from its native promoter on a plasmid in the Δ*rogA* strain restored survival to wild-type levels. This long-term survival benefit of the Δ*rogA* mutant was lost if the GTA locus was also deleted, demonstrating that the phenotype observed was GTA dependent (Fig 5A and 5B).

If the GTAs produced by Δ*rogA* cells promote survival, then this benefit should be cell-non-autonomous and therefore diffusible to other cells in the population. To test this idea, we mixed differentially marked wild-type and Δ*rogA*::$tet^R$ mutant cells. This co-incubation increased the viability of wild-type cells approximately 10-fold after 96 h in stationary phase relative to wild-type cells grown identically but alone (Fig 5C). There was also a corresponding

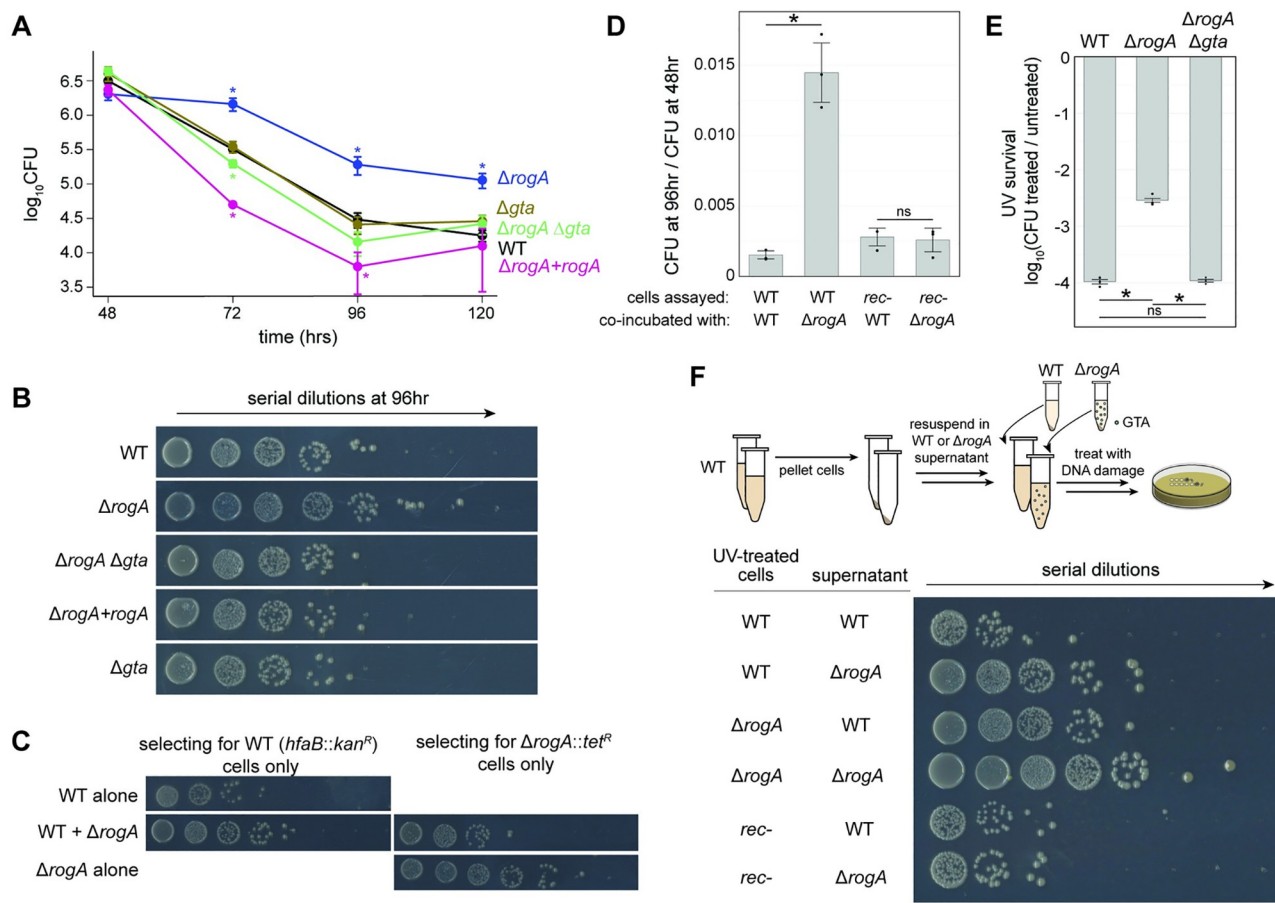

**Fig 5. GTAs provide a long-term survival benefit.** (A) Survival of various strains in PYE over several days ($n \geq 3$, error bars indicate SD). * = p-value <0.05 comparing data of same time point to wild type. Data are available in S1 Data. (B) Representative images of cell survival after 96 h in PYE. (C) Survival of wild-type (*hfaB::kanR*) or *ΔrogA::tet^R* cells when grown together or alone. Cells were enumerated on plates containing either kanamycin or tetracycline to select for either wild-type or *ΔrogA* cells. (D) Survival of the wild type (*hfaB::kan^R*) or the recombination-deficient *rec526* strain in combination with either wild-type or *ΔrogA* cells in stationary phase ($n = 3$, error bars indicate SD). * = p-value <0.05 between indicated data. ns = not significant. Data are available in S1 Data. (E) Survival of different strains in response to high levels of UV during stationary phase. Survival rates are plotted as the (CFU after treatment)/(CFU before treatment) ($n = 3$, error bars indicate SD). * = p-value <0.05 between indicated data. ns = not significant. Data are available in S1 Data. (F) Schematic for treating cells with UV after providing supernatants from different cultures. Survival of different strains with supernatant from either wild-type or *ΔrogA* stationary phase cultures. CFU, colony-forming unit; GTA, gene transfer agent.

decrease in survival of the *ΔrogA* cells when grown with wild-type cells, suggesting that the wild-type cells may act as "cheaters" and receive the benefit from GTA producers with no cost (Fig 5C). These findings support the notion that GTAs act as public goods and can benefit non-producing cells in the population. Given that we identified GTAs by studying DriD and the SOS-independent DNA damage response, we tested whether this benefit required DriD. We found that *ΔrogA* cells also bearing a deletion in *driD* still provided a benefit to co-incubated cells, suggesting the SOS-independent DNA damage response may not play a direct role in GTA production (S6A Fig).

We considered 3 potential models to explain the beneficial effects of GTA production on the viability of *C. crescentus* in stationary phase: (1) GTAs allow cells to acquire and spread new (relative to the wild-type genome), beneficial alleles; (2) GTA production causes cell lysis, which causes subsequent release of beneficial nutrients and DNA from the cytoplasm into the supernatant; or (3) GTAs allow cells to acquire and use wild-type DNA to repair or revert deleterious mutations.

To address the first model, we tested if the surviving wild-type cells following co-incubation with Δ*rogA* cells (Fig 5C) had acquired beneficial mutations from the Δ*rogA* strain and could better survive extended stationary phase. In many bacteria, mutations arise in stationary phase that provide a growth advantage; such mutations are heritable [27,28]. In principle, GTAs could be promoting the observed, approximately 10-fold increase in survival of wild-type cells co-incubated with the Δ*rogA* strain by driving distribution of such beneficial mutations. However, of 12 colonies tested here, all exhibited long-term survival comparable to naïve wild-type cells that had not been co-incubated with GTA producers (S6B Fig). This finding argued against model 1. Additionally, model 1 would have required any beneficial allele that arose in the GTA producer to somehow spread to most other cells in the population, which was unlikely given the timescale (72 to 96 h) of our experiments.

To distinguish between models 2 and 3, we first asked whether the survival benefit conferred by GTA-producing cells was recombination dependent. If the benefit results from nutrient release by lysed cells or direct delivery of nutrients by GTAs, it should be recombination independent. However, if the benefit involves GTA-mediated transfer of genetic material, it would require recombination. We used a recombination-deficient strain, *rec526*, that grows comparably to the wild type but is recombination deficient [29]. Whole-genome sequencing of the *rec526* strain revealed a truncation in *recA* due to a premature nonsense mutation (Q129*). When co-incubated for 96 h with either Δ*rogA* cells or wild-type cells, we observed no significant difference in the survival of the *rec526* cells (Fig 5D). This result suggested that the diffusible benefit from GTA-producing cells requires recombination and that GTAs may provide a template for recombination-dependent DNA repair.

## GTAs promote survival following DNA damage

The role of DNA uptake via competence machinery for DNA repair has been studied across many species [30]. To further test whether GTAs provide templates for DNA repair to promote survival following DNA damage, we exposed stationary phase cultures of different strains to UV light. Intriguingly, we observed a 21 ± 4-fold increase in survival of the Δ*rogA* strain, compared to wild-type cells (Fig 5E). This benefit was completely ablated if the GTA-encoding locus was also deleted. The increased survival of Δ*rogA* cells was only observed for stationary-phase cells with wild-type, Δ*rogA*, and Δ*rogA* Δ*gta* strains all exhibiting equivalent survival following UV treatment in exponential phase (S6C Fig). We also observed approximately 10-fold increase in the survival of UV-damaged wild-type cells exposed to GTA-rich supernatant from the Δ*rogA* strain (Fig 5F). Conversely, there was approximately 10-fold decrease in the survival of UV-damaged Δ*rogA* cells that recovered in GTA-poor supernatant from wild-type cells compared to recovering in Δ*rogA* supernatant. GTA-rich supernatant from Δ*rogA* cells provided no benefit to UV-damaged *rec526* cells, further supporting the conclusion that the benefit of GTAs is recombination dependent (Fig 5F).

We next tested whether DNA from GTAs can be used by cells to repair a specific DNA lesion, such as a double-strand break. For these experiments, we used a strain harboring both a vanillate-inducible, site-specific restriction enzyme (I-SceI) and a corresponding single restriction site on the chromosome [31]. Induction of this I-SceI system leads to a lethal double-strand break on the chromosome, resulting in sensitivity to vanillate. The strain also bore a chloramphenicol resistance marker to facilitate selection. This strain was grown to stationary phase and then co-incubated with an inducible GTA-producing "donor" strain (or a control strain) with the wild-type sequence in place of the I-SceI site (Fig 6A, top). Inducers were added to trigger the double-strand break in the recipient and GTA production in the donor. We then selected against the donor strain by plating on chloramphenicol while also either

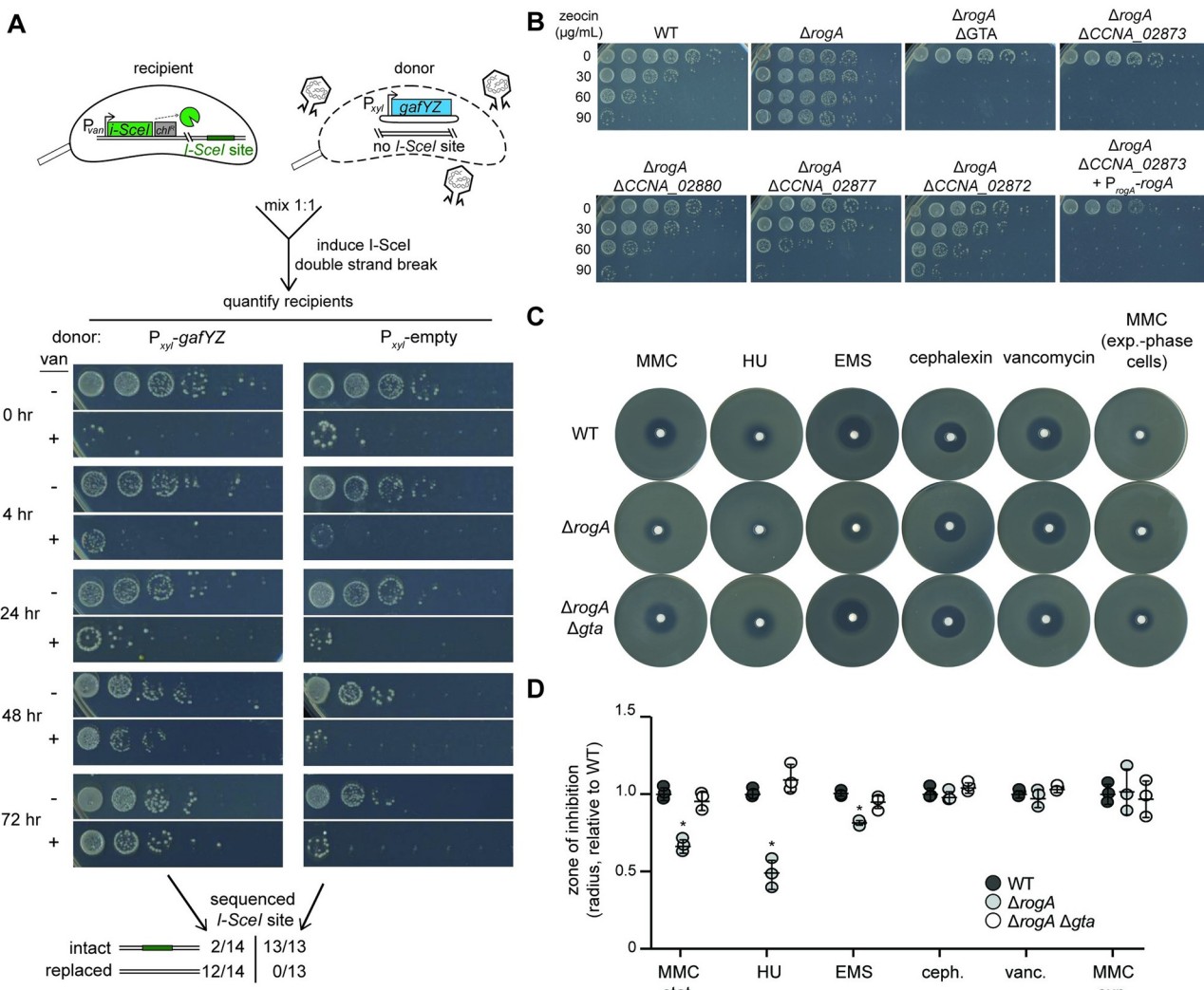

**Fig 6. GTAs confer survival to DNA damage.** (A) Survival of a recipient strain bearing the vanillate-inducible restriction enzyme I-SceI and an I-SceI site on the chromosome co-incubated with either a GTA producer (P$_{xyl}$-*gafYZ*) or a GTA non-producer control (P$_{xyl}$-empty). Vanillate was added to induce double-strand breaks in the recipient strain and xylose was added to induce the xylose promoter in the donors. Survival of the recipient strains was assayed after the indicated times with 10-fold serial dilutions on plates containing chloramphenicol with no vanillate (− van) to quantify all recipient cells and chloramphenicol with vanillate (+ van) to quantify all recipient cells that have lost the lethality of I-SceI expression. Outcome of sequencing the region harboring the I-SceI site in individual survivors is shown at the bottom. (B) Wild-type and various mutants were grown in PYE for 48 h and then exposed to a range of zeocin concentrations (0–90 μg/mL) for 24 h. Survival was enumerated with 10-fold serial dilutions. (C) Susceptibilities of different strains to the DNA-damaging agents MMC (0.5 mg/mL), HU (300 mg/mL), EMS (100 mg/mL), 2 control antibiotics cephalexin (6 mg/mL), and vancomycin (100 mg/mL). All cells tested were in stationary phase when plated except when indicated for MMC exp., where cells were in exponential phase. (D) Radius of inhibition of different strains to DNA-damaging agents and 2 control antibiotics. Radii are normalized to the WT mean radius for the drug (*n* = 3, error bars indicate SD). All cells tested were in stationary phase when plated except when indicated for MMC exp, where cells were in exponential phase. * = *p*-value <0.05 between indicated data and wild type. Data are available in S1 Data. GTA, gene transfer agent.

including vanillate, to further induce a double-strand break, or not. If GTAs mediate transfer of wild-type DNA that can be used to repair the break and thereby eliminate the I-SceI site, we would expect greater survival on vanillate for the recipient strain when incubated with a GTA-producing strain than the control strain. Only about $1 \times 10^{-3}$ cells were vanillate resistant after 4 h of co-incubation with GTA-producing or control donor cells, indicating that most cells still had an intact I-SceI site and the double-strand break was lethal, as expected (Fig 6A, t = 4

h). However, the recipient strain grown with GTA-producing donors had approximately 100-fold more vanillate-resistant colonies after 48 to 72 h than when grown with the control strain (Fig 6A). Sequencing confirmed that most (12 of 14) vanillate-resistant colonies from the culture with a GTA-producing donor had indeed replaced the I-SceI site with wild-type DNA (Fig 6A). In contrast, none (of 13) vanillate-resistant colonies from the culture with a control donor strain had lost the I-SceI site. We conclude that recipient cells can use the DNA from GTAs to repair their chromosomes. The frequency of transfer and integration of wild-type DNA that replaces the I-SceI site is likely rare, but those rare recombinants can then grow unabated in the presence of vanillate.

We also tested whether GTAs promote survival following treatment with the double-strand break agent zeocin (Fig 6B). We observed a substantial increase in survival of the Δ*rogA* strain compared to the wild type. This increased survival was GTA dependent as deleting any of 3 genes within the major GTA locus (*CCNA_02880* –terminase; *CCNA_02877* –capsid portal; *CCNA_02872* –major capsid) in the Δ*rogA* strain restored zeocin survival to approximately wild-type levels. A full deletion of the GTA locus actually produced hypersensitivity to zeocin suggesting that a gene within the GTA locus promotes resistance to double-strand breaks. Indeed, deleting the gene *CCNA_02873*, which is on the antisense strand of the GTA locus and annotated as a putative bleomycin resistance gene, was responsible for the hypersensitivity of the Δ*rogA* Δ*gta* strain. Complementation of *rogA* in the Δ*rogA* Δ*CCNA_02873* strain did not restore sensitivity to wild-type levels, further suggesting that loss of *CCNA_02873* alone was responsible for the increased zeocin sensitivity. Collectively, these results are consistent with the notion that GTAs may promote survival to double-strand breaks.

To test a more general role of GTAs in mediating survival after DNA damage, we treated wild-type, Δ*rogA*, and Δ*rogA* Δ*gta* strains with 3 other DNA-damaging agents (mitomycin C, MMC; hydroxyurea, HU; and ethyl methanesulfonate, EMS) and 2 non-DNA-damaging antibiotics (cephalexin and vancomycin) using disk diffusion assays. A small disk of filter paper impregnated with a given drug was placed on a hard agar plate with a top layer of soft agar containing cells taken from stationary phase cultures. The resulting inhibition zone radius is inversely proportional to the survival of the strain in the presence of the drug. The Δ*rogA* strain survived better than the wild-type or the Δ*rogA* Δ*gta* strain in the presence of MMC and HU (Fig 6C and 6D), which can induce double-strand breaks directly or via replication fork collapse, respectively. For MMC, we found no difference between the different strains if the cells plated came from exponential phase cultures (Fig 6C and 6D). There was also a small increase in survival against EMS, which causes DNA damage (base alkylation) that is repaired by mismatch or base excision repair, but can cause double-strand breaks at high levels [32]. In contrast to MMC, HU, and EMS, no significant difference in survival was detected for Δ*rogA* cells in the presence of cephalexin or vancomycin, which inhibit cell growth independent of DNA damage (Fig 6C and 6D). Taken together, our results suggest that GTAs can improve a recipient cell's ability to survive DNA damage.

## Discussion

### Physiological benefit of GTA production

GTAs are intriguing, but still poorly understood, vehicles for HGT, particularly in α-proteobacteria. *Caulobacter crescentus* was not previously known to produce GTAs and bioinformatic studies even suggested it may not produce functional particles [9]. However, our work indicates that *Caulobacter* can indeed produce GTAs. Moreover, these GTAs, each harboring approximately 8.3-kb fragment of the genome, can transfer genetic material to recipient cells. Enrichment of packaging was seen around the 1.0 and 3.0 Mbp positions, which result from

chromosomal arm cohesion, localization of those regions to mid-cell, and proximity to the GTA cluster [33]. The DNA transferred can enable cells to acquire beneficial alleles or wild-type DNA that can be used for recombination-based repair (Fig 7).

The physiological benefits of GTA production have long been a mystery in the field. GTAs can transfer beneficial alleles, such as antibiotic resistance genes, that aid survival during a subsequent selection imposed in the lab, but whether a similar scenario arises in nature is unknown. Here, we presented evidence that GTA production may provide a long-term survival benefit to *C. crescentus*. A GTA-producing population of Δ*rogA* cells survived long-term stationary phase approximately 10-fold better than wild-type cells in a GTA-dependent manner. This benefit appears to be a public good as co-incubating a GTA-producer with a non-producer increased survival of the latter. However, the benefit was not the transfer of a beneficial allele as the wild-type cells that survived better in the presence of Δ*rogA* cells behaved like naïve, monocultured wild-type cells in a second round of stationary phase challenge (S6 Fig).

Importantly, the benefit of GTAs to recipient cells depends on homologous recombination. This result suggests that the DNA provided by GTAs is not used simply as a nutrient source and instead must be integrated into the genome. We found that GTA-producing cells better survived UV-induced DNA damage in stationary phase, again in a recombination-dependent manner. Thus, our results suggest that GTA production confers a benefit to *C. crescentus* during long-term growth conditions, potentially by acting as a reservoir of DNA for the repair of damaged chromosomal DNA. Consistent with this model, we found that GTA production was less advantageous when cells were challenged with other forms of DNA damage, such as alkylating agents, that can be repaired through pathways other than homologous recombination. UV damage, which typically leads to interstrand crosslinks, depends on homologous recombination to be repaired. Moreover, we found that GTAs were beneficial to a population of cells in which we introduced a single, site-specific double-strand break, a lesion that also requires homologous recombination for repair. *C. crescentus* does not encode a non-homologous end-joining system like some bacteria [34]. A hallmark of *C. crescentus*, and many α-proteobacteria, is that they often harbor only a single chromosome—particularly in stationary phase [35] —meaning they lack a sister chromosome template for homologous recombination-based repair. GTAs may represent a solution to this problem.

Whether providing cells with templates for DNA repair is a bona fide function of GTAs in wild populations of cells remains to be shown. The role of DNA uptake during competence, or natural transformation, has been studied and reviewed at length [36], with several potential models suggested: The DNA is used as a nutrient source [37], for DNA repair [30,38], or for increasing genetic diversity [39]. Notably, our results suggest GTAs act to benefit cells by providing a source of DNA during DNA-damaging conditions, in line with competence studies suggesting a role in DNA repair. One caveat of our model is that the incoming DNA includes only approximately 8-kb region of the genome. However, given large population sizes and the potentially large "burst size" from GTA producers, repair could still occur at a high enough frequency to explain our observations that damaged wild-type cells benefit from co-incubation with GTA producers and can use GTAs to repair specific lesions (Fig 6A). Increasing recombination within a population via GTA production, especially in a species like *C. crescentus* that lacks competence machinery, may also help to avoid the accumulation of deleterious mutations, a phenomenon known as Mueller's Ratchet [40].

Another evolutionary conundrum regarding GTA production is that the benefit of GTAs must outweigh the heavy cost of producing GTAs, which requires cell lysis and the altruistic sacrifice of some cells for others in the population. Unlike natural transformation, where the production of uptake machinery occurs in the receiving cell and thus any benefit of transformation could outweigh the cost of machinery production, the energetic burden of GTA

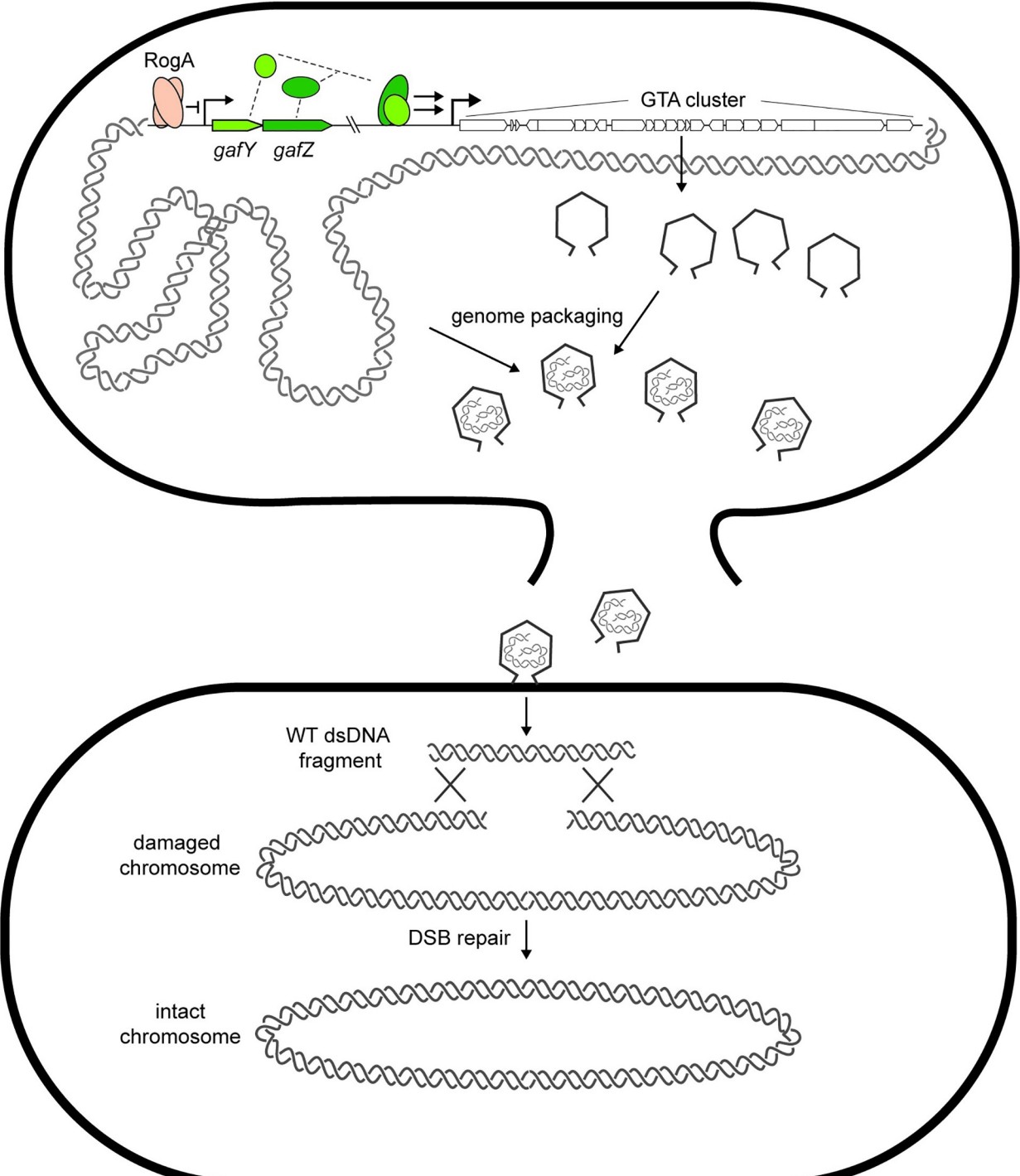

**Fig 7. Model schematic of GTA production in *Caulobacter crescentus*.** RogA directly represses expression from the *gafYZ* promoter. From some unknown stimulus, RogA is removed from the promoter, leading to transcriptional activation of *gafYZ*, which are sufficient to induce expression of the GTA cluster and other auxiliary GTA genes. GTA induction leads to production of phage-like particles that are packed with random fragments of genomic DNA. The producer cell then lyses and releases GTA particles into the supernatant that can then transfer wild-type DNA to recipient cells to be used for recombination to fix deleterious mutations or strand breaks. GTA, gene transfer agent.

production appears to be solely on the producer, which will then go on to lyse and guarantee cell death. This evolutionary constraint necessitates a population-wide benefit to outweigh the cost of programmed cell death in individuals, such as seen in division of labor models or social evolution theory [41,42]. There are some examples of direct altruism in bacteria, including phage abortive infection, in which infected cells in a population die to prevent phage propagation and thereby save uninfected neighbor cells [43], and biofilm development, where cell lysis releases structural components needed to make a robust biofilm matrix [44]. However, the benefit to neighboring cells is less clear for GTAs produced by bacteria like *Caulobacter* in aqueous, high-diffusion environments where particles may be easily washed away from clonal neighbors. To counter this, GTA production may be linked to high cell density to promote transfer, as seen in this work and other GTA systems [16,45,46]. It may also be necessary for cells to ensure that any GTAs released can only be received by a closely related cell, e.g., a dedicated cell receptor/tail-fiber pair could restrict the benefit only to other cells of the same species as is seen in *R. capsulatus* [16]. In this way, a recipient cell could be informed of the incoming DNA's origin, similar to the role of uptake sequences in transformation [47,48]. Further studies investigating whether the GTAs of *C. crescentus* can transfer DNA to closely related species will shed light on species-specificity and potential GTA receptors. The lack of apparent tails of the GTA in *C. crescentus* despite encoding tail proteins is surprising, but may be a result of physical forces during purification having disrupted tail structures. Further work to purify GTAs with other methods or visualization of GTAs in vivo with cryo-electron tomography, as well as testing which genes in the GTA cluster are required for transfer may help resolve this issue. Finally, if the benefit of receiving GTAs requires recombination, and thus relies on high sequence similarity, the benefit would also be restricted primarily to closely related strains and species.

## Regulation of GTA production

Not surprisingly, GTA biogenesis is a tightly regulated process as it ultimately leads to lysis of the producing cell. Recent work in *R. capsulatus* identified GafA as a direct transcriptional activator of GTA production [21], although the mechanism of activation is not known. In *Caulobacter*, and likely many other α-proteobacteria, GafY and GafZ appear to be a split, 2-gene homolog of GafA. As with GafA in *R. capsulatus*, GafY and GafZ are necessary and, when expressed together, sufficient for GTA production in *Caulobacter*. Although GafYZ form a heterocomplex, as evidenced by co-IP studies, our ChIP-seq analysis indicated that GafY alone binds the promoter regions of many operons, including the long, primary GTA locus, whereas GafZ was found only within the GTA locus. The binding pattern of GafZ suggests that it may travel with and possibly regulate RNA polymerase during the transcription of this 21-gene cluster.

GafY (locus *CCNA_01111* in *C. crescentus*) is homologous to the DNA-binding domain of DnaA (domain IV). A prior study found that this gene is induced in stationary phase and that overexpressing it increased the copy number of some genomic loci, inhibited cell division, and eventually led to cell death [22]. These observations are consistent with ours and each can be explained by the impact of GafY on GTA production documented here. Thus, we propose that GafY does not regulate *oriC*-based chromosome replication via DnaA and instead functions in GTA biogenesis.

We also identified the transcription factor RogA as a direct repressor of GafYZ and ultimately GTA production. RogA homologs are found throughout the α-proteobacteria and are particularly enriched in the *Rhizobiales*, *Rhodospirillales*, *Rhodobacterales*, and *Caulobacterales*, where they may similarly control *gafYZ* or *gafA* homologs to control GTA production. RogA

has a domain organization similar to LexA and λ cI, with an N-terminal helix-turn-helix domain and an S24 auto-peptidase-like domain, though lacks the key catalytic residues.

Notably, we had identified *rogA* in a screen for regulators of *didA*, which encodes a DNA damage-inducible cell division inhibitor. The expression of *didA* is directly controlled by the transcription factor DriD, with strongest induction triggered by double-strand breaks [18]. Our work here indicates that RogA directly controls *gafYZ*, which then indirectly stimulate *didA* expression. In *rogA* mutants, *gafYZ* are induced, leading to induction of GTA synthesis. The assembly of GTA particles requires the packaging of 8.3-kb linear fragments of the genome, which then trigger DriD to activate *didA* and other DriD targets. It is not yet clear whether the induction of *didA* and its ability to inhibit cell division is somehow advantageous to GTA producer cells.

Although RogA and GafYZ (or GafA) appear to be relatively common, conserved regulators of GTA synthesis, there may also be species- or clade-specific regulators. For example, the CckA-ChpT-CtrA phosphorelay and quorum-sensing proteins GtaI and GtaR regulate GTA synthesis in *Rhodobacter* [49–53], but not in *Caulobacter* [54]. The stringent response, mediated by ppGpp, also somehow promotes GTA synthesis in *Rhodobacter* [45].

In both *Caulobacter* and *Rhodobacter*, GTA synthesis occurs almost exclusively in stationary phase. Although cells can receive GTAs in either exponential or stationary phase, synthesis and subsequent release is restricted to stationary phase. However, entry to stationary phase is necessary, but not sufficient. To detect GTA synthesis, we had to delete *rogA* or overexpress *gafYZ*. There are likely external signals or cues that trigger GTA synthesis in wild-type cells, possibly by inhibiting RogA or by stimulating GafYZ activity. DNA-damaging agents are one prime candidate given our finding that GTAs can provide a template for homologous recombination-based DNA repair. Why GTA synthesis is restricted to stationary phase warrants further investigation. It may be that an additional signal is normally coincident with or also requires stationary phase. For example, in *Rhodobacter*, GTA production is under quorum-sensing control, possibly to ensure that released GTAs will efficiently find recipient cells.

## Concluding remarks

The prevalence and conservation of GTAs, particularly in α-proteobacteria, has long suggested that they provide cells a benefit, but the nature of this benefit has remained enigmatic. Our work now points to a possible role for GTAs in promoting DNA repair by providing recipient cells with templates for homologous recombination. However, we cannot rule out a role in facilitating the acquisition of beneficial alleles in some conditions or specific scenarios. Whatever the case, our discovery that *C. crescentus* produces functional GTAs also now offers a new, highly tractable organism to further dissect the functions, biogenesis, and regulation of GTAs.

## Methods

Plasmids and strains used in this study are listed in S1 and S2 Tables, respectively. DNA oligonucleotides used in strain construction and experiments are listed in S3 Table. For statistical tests, 2-tailed *t* tests were used with a significance cutoff of *p*-value <0.05.

## Plasmids

*C. crescentus* CB15N genomic DNA was used as template for PCR amplifications unless otherwise noted. All PCR products were digested with noted restriction enzymes and ligated into the double-enzyme cut corresponding plasmid. Approximately 5 μL of ligation reactions were

subsequently transformed into chemically competent *E. coli* DH5α cells. All resulting plasmids were verified by Sanger sequencing (Genewiz).

To generate, pBXMCS-2::P$_{xyl}$-*gafY*, pBXMCS-2::P$_{xyl}$-*gafZ*, and pBXMCS-2::P$_{xyl}$-*gafYZ*, the coding sequence of *gafY*, *gafZ*, or the native operon of *gafYZ* were PCR-amplified with primers oKRG92 and oKRG93, oKRG400 and oKRG401, or oKRG92 and oKRG401, respectively. Amplicons were digested with NdeI and SacI and ligated into cut pBXMCS-2.

To generate, pRVMCS-2::P$_{rogA}$-*rogA*, the promoter and coding sequence of *rogA* was PCR-amplified with primers oKRG127 and oKRG128, digested with SacII and AflII and ligated into cut pRVMCS-2.

To generate pNPTS138::*tet*$^R$ *1.0Mb*, a *tet*$^R$ cassette was cloned with approximately 600-bp flanking sites near the 1.0 Mb position as follows: Approximately 600-bp region including part of the coding region of *CCNA_00928* up to the intergenic region to *CCNA_00929* was amplified by PCR using primers oKRG446 and oKRG447. Approximately 600-bp region consisting of the *CCNA_00928-CCNA_00929* intergenic region and some of the *CCNA_00929* coding region was amplified by PCR using primers oKRG448 and oKRG449. The tetracycline resistance cassette was amplified from the plasmid pMCS-5 [55] using primers oKRG440 and oKRG441. These 3 PCR products were fused using fusion PCR with primers oKRG446 and oKRG449, gel purified, digested with EcoRI and SalI and ligated into pNPTS138.

To generate pNPTS138::*tet*$^R$ *2.0Mb*, an identical approach was used as for pNPTS138::*tet*$^R$ 1.0 Mb, instead with the *tet*$^R$ cassette cloned with approximately 600-bp flanking sites near the 2.0 Mb position. The regions upstream and downstream of the *CCNA_01861-CCNA_01862* intergenic region were amplified using primers oKRG458 with oKRG459 and oKRG460 with oKRG461, respectively. Fusion PCR with the *tet*$^R$ cassette and ligation into pNPTS138 was performed as described for pNPTS138::*tet*$^R$ *1.0Mb*.

To generate pNPTS138::Δ*rogA*, a 600-bp region upstream of *rogA* that includes the first 9 nucleotides of *rogA* was amplified by PCR using primers oKRG102 and oKRG103. A 600-bp downstream of *rogA* that includes the last 9 nucleotides of *rogA* was amplified by PCR using primers oKRG104 and oKRG105. These 2 PCR products were fused using fusion PCR with primers oKRG102 and oKRG105, gel purified, digested, and ligated into a SpeI-AflII-cut pNPTS138.

To generate pNPTS138::Δ*rogA*::*tet*$^R$, a 600-bp region upstream of *rogA* that includes the first 9 nucleotides of *rogA* was amplified by PCR using primers oKRG102 and oKRG106. A 600-bp downstream of *rogA* that includes the last 9 nucleotides of *rogA* was amplified by PCR using primers oKRG107 and oKRG105. The tetracycline resistance cassette was amplified by PCR using primers oKRG440 and oKRG441 with the previously described pMCS-5 [55] as template. These 3 PCR products were fused using fusion PCR with primers oKRG102 and oKRG105, gel purified, digested, and ligated into a SpeI-AflII-cut pNPTS138.

To generate pNPTS138::Δ*gta*, a 600-bp region including 300-bp upstream of *CCNA_02880* and 300 bp into the coding region of *CCNA_02880* was PCR amplified using primers oKRG111 and oKRG110. A 600-bp downstream of *CCNA_02861* that includes the last 300 nucleotides of *CCNA_02861* was amplified by PCR using primers oKRG109 and oKRG108. These 2 PCR products were fused using fusion PCR with primers oKRG108 and oKRG111, gel purified, digested, and ligated into a EcoRI-cut pNPTS138.

To generate pNPTS138::*hfaB*::*kan*$^R$, upstream half and downstream half of *hfaB* were PCR amplified using primers oKRG114/oKRG115 and oKRG112/oKRG113. The kanamycin resistance cassette was PCR amplified with primers oKRG116 and oKRG117 and pBXMCS-2 as template. These 3 PCR products were fused using fusion PCR with primers oKRG112 and oKRG115, gel purified, digested, and ligated into a SpeI-AflII-cut pNPTS138.

To generate pNPTS138::Δ*gafY*, a 500-bp region upstream of *gafY* that includes the first 9 nucleotides of *gafY* was amplified by PCR using primers oKRG118 and oKRG119. A 500-bp

downstream of *gafY* that includes the last 9 nucleotides of *gafY* was amplified by PCR using primers oKRG120 and oKRG121. These 2 PCR products were fused using fusion PCR with primers oKRG118 and oKRG121, gel purified, digested, and ligated into a EcoRI-cut pNPTS138.

To generate pNPTS138::Δ*gafZ*, a 500-bp region upstream of *gafZ* that includes the first 9 nucleotides of *gafY* was amplified by PCR using primers oKRG122 and oKRG123. A 500-bp downstream of *gafY* that includes the last 9 nucleotides of *gafY* was amplified by PCR using primers oKRG124 and oKRG125. These 2 PCR products were fused using fusion PCR with primers oKRG122 and oKRG125, gel purified, digested, and ligated into a SpeI-AflII-cut pNPTS138.

To generate pNPTS138::Δ*gafYZ*, a 500-bp region upstream of *gafY* that includes the first 9 nucleotides of *gafY* was amplified by PCR using primers oKRG118 and oKRG119. A 500-bp downstream of *gafZ* that includes the last 9 nucleotides of *gafY* was amplified by PCR using primers oKRG126 and oKRG125. These 2 PCR products were fused using fusion PCR with primers oKRG118 and oKRG125, gel purified, digested, and ligated into a SpeI-AflII-cut pNPTS138.

To generate pNPTS138::Δ*rogA*::*tet*$^R$ 2 (maintaining a longer stretch of the starting nucleotides of the open reading frame), a 500-bp region upstream of *rogA* that includes the first 30 nucleotides of *rogA* was amplified by PCR using primers NTP2194 and NTP2305 and a 500-bp downstream of *rogA* that includes the last 9 nucleotides of *rogA* was amplified by PCR using primers NTP2306 and NTP2197. The tetracycline resistance cassette was amplified by PCR using primers NTP2303 and NTP2304, and as template the previously described pMCS-5 [55]. All 3 resulting PCR products were gel purified and assembled together into BamHI-HindIII-cut pNPTS138 using 2× Gibson mastermix.

To generate pNPTS138::Δ*CCNA_02880*, pNPTS138::Δ*CCNA_02877*, pNPTS138:: Δ*CCNA0041_02872*, pNPTS138::Δ*CCNA_02873*, approximately 500-bp upstream region of each gene including 3 to 5 codons into the coding region was amplified using the primer pairs NTP2230 and NTP2231, NTP2493 and NTP2494, NTP2222 and NTP2223, or NTP2323 and NTP2324, respectively. The downstream region of each gene was amplified by PCR using primer pairs NTP2232 and NTP2233, NTP2495 and NTP2496, NTP2224 and NTP2225, or NTP2325 and NTP2326, respectively. Each upstream and downstream PCR products were gel purified and assembled together into BamHI-HindIII-cut pNPTS138 using 2× Gibson mastermix.

To generate pENTR::*gafY*, the coding sequence of GafY was PCR-amplified using primers NTP2568 and NTP2569. The resulting PCR product were gel purified and assembled with the pENTR plasmid backbone using 2× Gibson mastermix (NEB).

To generate pML333::*his*$_6$-*mbp*-*gafY*, DNA containing *gafY* was recombined into a Gateway-compatible destination vector pML333 via an LR recombination reaction (Invitrogen). For LR recombination reactions: 1 μL of purified pENTR::*gafY* was incubated with 1 μL of pML333, 1 μL of LR Clonase II mastermix (Invitrogen), and 2 μL of water in a total volume of 5 μL. The reaction was incubated for an hour at room temperature before being introduced into *E. coli* DH5α cells by heat-shock transformation. Cells were then plated out on LB agar + carbenicillin. Resulting colonies were restruck onto LB agar + carbenicillin and LB agar + kanamycin. Only colonies that formed on LB + carbenicillin plates were used for culturing and plasmid extraction.

To generate *pET21b::rogA-his*$_6$, the coding sequence of RogA was PCR amplified using primers NTP2291 and NTP2292. The resulting PCR product was gel purified and assembled into an NdeI-HindIII-cut pET21b using a 2× Gibson mastermix.

To generate pCOLADuet-1:*his₆-gafZ*, the coding sequence of GafZ was PCR amplified using primers NTP2693 and NTP2694. The resulting PCR product was gel purified and assembled into an EcoRI-HindIII-cut pCOLADuet-1 plasmid backbone using a 2× Gibson mastermix.

To generate pCOLADuet-1:*his₆-gafZ gafY*, the coding sequence of (tagless) GafY was PCR amplified using primers NTP2691 and NTP2692. The resulting PCR product was gel purified and assembled into an NdeI-KpnI-cut pCOLADuet-1:*his₆-gafZ* plasmid backbone using a 2× Gibson mastermix.

To generate pNPTS138::*flag-gafZ*, a 500-bp region upstream of the FLAG-tag insertion site was amplified by PCR using primers NTP2598 and NTP2599. A 500-bp downstream of the FLAG-tag insertion site was amplified by PCR using primers NTP2600 and NTP2601. The resulting PCR products were gel purified and assembled together into a BamHI-HindIII-cut pNPTS138 using a 2× Gibson mastermix.

To generate pNPTS138::Δ*gafY*::*8xTAG*, a 500-bp region upstream of the 8xstop codon (TAG) insertion site was amplified by PCR using primers NTP2642 and NTP2643. A 500-bp downstream of the 8xTAG insertion site was amplified by PCR using primers NTP2644 and NTP2645. The resulting PCR products were gel purified and assembled together into a Bam-HI-HindIII-cut pNPTS138 using a 2× Gibson mastermix.

## Strains

For strain construction, insertions and deletions were verified by PCR using primers outside the genetic alteration, as appropriate. Double-crossover gene knock-ins were performed with 2-step recombination as described previously [56]. Transcriptional reporters (P*didA-lacZ*) and antibiotic cassette markers (*kan^R*) were inserted at the *hfaB* locus, which encodes for holdfast protein and is a dispensable region in CB15N [57] commonly used for genomic insertions.

To generate ML3660 and ML3661, the insert in the plasmid pNPTS138::Δ*rogA*::*tet^R* was introduced by 2-step recombination to CB15N and ML2169, respectively.

To generate ML3662, the insert in the plasmid pNPT-spec-Δ*driD* was introduced by 2-step recombination to ML3660.

To generate ML3663 and ML3664, the plasmid pRVMCS-2::P*rogA-rogA* was introduced by electroporation into ML3661 and ML3660, respectively.

To generate ML3665, ML3666, ML3667, ML3668, and ML3669, the inserts in plasmids pNPT-spec-Δ*driD*, pNPTS138::Δ*gta*::*kanR*, pNPTS138::Δ*gafY*, pNPTS138::Δ*gafZ*, and pNPTS138::Δ*gafYZ* were introduced into ML3660 by 2-step recombination, respectively.

To generate ML3670, ML3671, ML3672, and ML3673, the plasmids pBXMCS-2 (empty), pBXMCS-2::P*xyl-gafY*, pBXMCS-2::P*xyl-gafZ*, and pBXMCS-2::P*xyl-gafYZ* were introduced into CB15N by electroporation.

To generate ML3674 and ML3675, the inserts in plasmids pNPTS138::Δ*rogA* and pNPTS138::Δ*gta* were introduced by 2-step recombination to CB15N, respectively.

To generate ML3676, the plasmid pBXMCS-2::P*xyl-gafYZ* was introduced by electroporation into ML3675.

To generate ML3677 and ML3678, the plasmids pNPTS138::*tet^R 1.0Mb* and pNPTS138::*tet^R 2.0 Mb* were first introduced by 2-step recombination to CB15N, respectively. The plasmid pBXMCS-2::P*xyl-gafYZ* was then subsequently introduced to each of these intermediate strains to generate ML3677 and ML3678.

To generate ML3679, the insert in plasmid pNPTS138::*hfaB*::*kan^R* was introduced by 2-step recombination to CB15N. Then, the plasmid pXGFPC-6 was integrated into this strain at the xylose promoter by 1-step recombination.

To generate ML3680, an I-SceI restriction cut site located near the CCNA_00727 locus was introduced into ML2169 by ΦCr30-mediated transduction from the strain ML2466. The resulting strain was then used as the recipient strain for a second ΦCr30-mediated transduction of the *i-SceI* gene integrated at the vanillate locus and marked by a chloramphenicol resistance cassette from ML2466.

To generate ML3681, the plasmid pXGFPC-2 (empty) was introduced into ML3680 by electroporation.

To generate ML3682, ML3683, and ML3685, the plasmids pBXMCS-2 (empty), pBXMCS-2::P$_{xyl}$-*gafY*, and pBXMCS-2::P$_{xyl}$-*gafYZ* were introduced into ML2169 by electroporation.

To generate NTS2275, which bears a deletion of *rogA* with the 30 nucleotides of the ORF remaining, the insert in plasmid pNPTS138::Δ*rogA*::*tet*$^R$ 2 was introduced into CB15N by 2-step recombination.

To generate NTS2481, which contains a FLAG tag at the N-terminus of GafZ, the insert in the plasmid pNPTS138::*flag-gafZ* was introduced into CB15N by 2-step recombination.

To generate NTS2501, which bears an 8xstop codon (TAG) downstream of the start codon of *gafY* to disrupt *gafY* without affecting expression of *gafZ*, the plasmid pNPTS138::Δ*gafY*::*8x-TAG* was introduced into CB15N by 2-step recombination.

To generate NTS2515 and NTS2489, the *rogA*::*tet*$^R$ allele from NTS2275 was introduced into NTS2501 and NTS2481 by ΦCr30-mediated transduction, respectively.

To generate NTS2316, NTS2445, NTS2314, and NTS2403, the individual gene deletions from pNPTS138::Δ*CCNA_02880*, pNPTS138::Δ*CCNA_02877*, pNPTS138::Δ*CCNA_02872*, and pNPTS138::Δ*CCNA_02873* were introduced by 2-step recombination to CB15N, respectively. To each of the resulting intermediate strains, the *rogA*::*tet*$^R$ allele from NTS2275 was introduced by ΦCr30-mediated transduction to generate the respective double mutants.

## PCR conditions

PCR was performed with Phusion HF DNA polymerase using 5X Phusion GC Reaction Buffer (NEB). Each reaction contained 10 μL buffer, 10 μL 3M Betaine monohydrate (Sigma), 4 μL dNTPs, 5 μL of a 10 μm forward and reverse primer mix, 50 ng template, 1 μL DMSO, 0.5 μL polymerase, and nuclease-free water to 50 μL. Two-step cycling was performed as follows: 98˚C 30 s, 34× (98˚C 10 s, 72˚C 30 s/kb), 72˚C 5 min. Fusion PCR was performed similarly with 50 ng of the largest template fragment and equimolar amounts of the smaller template fragments and an additional annealing step of 20 s at 60˚C was added.

## Growth conditions

*Caulobacter* strains were grown at 30˚C in rich medium (PYE) shaking in flasks. To induce or repress expression from the P$_{xyl}$ promoter, liquid media was supplemented with 0.3% xylose or 0.2% glucose, respectively. To induce or repress expression from P$_{van}$, liquid media was supplemented with or without 500 μM vanillic acid (Fluka). Mitomycin C (A&G) was added to liquid cultures at 1 μg/mL and to agarose pads and agar plates at 0.35 μg/mL unless otherwise indicated. To induce expression from P$_{lac}$, plate media were supplemented with 75 or 100 μM IPTG (Sigma). Novobiocin (Fluka) and cephalexin (Sigma) were added to agar plates at 0.35 μg/mL and 7.5 μg/mL, respectively. Antibiotics were used at the following concentrations for strain construction and plasmid maintenance in *Caulobacter* cells in liquid:plates—kanamycin 5 μg/mL: 25 μg/mL; oxytetracycline 1 μg/mL: 2 μg/mL; chloramphenicol 2 μg/mL: 1 μg/mL; gentamycin 2.5 μg/mL: 5 μg/mL. Transformations and transductions were performed as previously described [58]. All cell survival and CFU enumeration assays were performed with 10-fold serial dilutions, unless otherwise noted.

## Tn-mutagenesis screen for identifying upstream regulators of *didA* expression

Cells expressing *lacZ* driven by the *didA* promoter integrated at the chromosomal *hfaB* locus were mutagenized with the EZ-TN5 transposome kit (Epicentre) and were selected for transposon insertion on PYE plates containing kanamycin and 20 μg/mL X-gal. Dark blue colonies were isolated and transposon insertions were identified using rescue cloning with *pir-116* electrocompetent *E. coli* cells (Epicentre, TSM08KR).

## β-galactosidase assay

β-galactosidase activity was measured in mid-log ($OD_{600}$ = 0.2–0.3) or stationary phase ($OD_{600}$ = 1.2–1.3). Strains bearing the $P_{didA}$-*didA* reporter were permeabilized by adding 100 μL of chloroform to 800 μL of cells followed by vortexing. Cells were incubated at 30˚C for 15 min prior to addition of ortho-nitrophenyl-β-galactoside. The assay and subsequent activity calculations were done as previously described [59].

## RNA-seq

RNA-seq was performed as previously described [60]. Briefly, RNA-seq libraries were sequenced by paired-end sequencing (75 nt) on an Illumina NextSeq500 sequencer at the MIT BioMicro Center. Custom scripts written in Python 2.7.6 were used to perform data analysis. The paired-end reads were mapped to *Caulobacter* NC011916.1 using bowtie2 with default parameters [61]. The read coverage was mapped by assigning each mapped base a value of $1/N$ where $N$ equals the length of each paired read. mRNA abundance was calculated by first adding a pseudocount to all positions and then the number of reads mapped to a gene was divided by the gene's length and normalized to yield the mean number of reads per kilobase per million sequencing reads (RPKM). Changes in gene expression were calculated by taking the $\log_2$ transformation of the ratio of the RPKM of each gene from the experimental condition ($P_{xyl}$-*gafYZ* in xylose or Δ*rogA*) to the respective control ($P_{xyl}$-*empty* in xylose or wild-type). RNA-seq data are available at GEO accession number GSE184480.

## Genomic DNA isolation

Total DNA extractions were performed by pelleting 1 mL of stationary phase cells. Pellets were resuspended in 600 μL Cell Lysis Solution (Qiagen) and lysed by incubation at 80˚C for 5 min. A total of 50 μg of RNAse A was then added to the cell lysate and incubated at 37˚C for 30 min to digest cellular RNAs. Proteins were precipitated from the lysate by adding 200 μL of Protein Precipitation Solution (Qiagen), vortexing and setting the sample on ice for 30 min, and pelleting for 10 min at 14,000 rpm. The supernatant was then mixed with 600 μL of isopropanol and inverted to precipitate the DNA. The DNA was pelleted via centrifugation at 14,000 rpm for 3 min, washed once with 70% ethanol, and resuspended in 100 μL of $H_2O$.

## PacBio DNA sequencing

To sequence the DNA packaged in GTA particles, cells bearing the $P_{xyl}$-*gafYZ* overexpression plasmid were induced with 0.3% xylose for 3 h after reaching an $OD_{600}$ of 1.3. Total DNA was then extracted as above. The GTA DNA was separated from chromosomal genomic DNA via gel electrophoresis in a 1% agarose gel and subsequent gel extraction (Zymoclean Gel DNA extraction kit). The DNA was further purified by isopropanol precipitation and an ethanol wash and was resuspended in Qiagen elution buffer. The resulting purified GTA DNA was sequenced using PacBio SMRT sequencing technology at the MIT BioMicro Center. Briefly,

the dsDNA library was prepared using the Express Template Prep Kit v2.0, consisting of end repair, DNA damage repair, adapter ligation, further purification, and quality control. The library was sequenced with a 10-h movie on a PacBio Sequel v3. Sequencing reads were analyzed using circular consensus sequencing. Full consensus reads were aligned to the *Caulobacter* NC011916.1 genome with Bowtie 2. PacBio sequencing data are available at GEO accession number GSE184478.

## GTA induction for CFU, lysis, and gene transfer assays

For induction of GTA expression, cells bearing $P_{xyl}$-*gafYZ* on a high-copy plasmid were grown up to stationary phase ($OD_{600}$ = 1.2–1.3) in PYE with 0.2% (w/v) glucose, unless otherwise noted. Xylose was added to a final concentration of 0.3% (w/v) to induce expression. Cells were then used for enumerating survival with CFU assays, DNA extractions to isolate GTA-packaged DNA, or as donor cells for subsequent gene transfer assays. To quantify cell lysis, cultures at each time point were pelleted by centrifugation and cell-free supernatant was assayed for protein content using the Pierce Coomassie Plus Assay Kit (Thermo), following manufacturer's protocol.

## DNA staining

Cells bearing the $P_{xyl}$-*gafYZ* plasmid were induced with 0.3% xylose for 3 h after reaching $OD_{600}$ = 1.3. Cells were pelleted, resuspended in PBS, and SYBR Gold (Invitrogen) was added at a 1:5,000 fold dilution from the stock. Cells were incubated at room temperature for 5 min. One microliter was spotted on 1.5% agarose in PBS pads and imaged with phase contrast and epifluorescence on a Zeiss Observer Z1 microscope using a 100×/1.4 oil immersion objective and an LED-based Colibri illumination system with MetaMorph software (Universal Imaging, PA). Dead cell staining with propidium iodide was performed similarly as SYBR Gold staining, with a final concentration of 1 μM propidium iodide.

## Gene transfer assay and transfer during a single double-strand break

Cultures of donor cells bearing the $P_{xyl}$-*gafYZ* plasmid and a $tet^R$ marker at the 1.0 Mbp locus, and recipient cells bearing the empty high-copy vector pBXMCS-2 and a chloramphenicol resistance cassette at the *hfaB* locus, were grown to $OD_{600}$ = 1.3 in PYE containing 0.2% glucose to repress the xylose promoter and kanamycin to maintain plasmid selection. The cultures were mixed 1:1, or at noted ratios, in 250 mL flasks and induced with 0.3% xylose for the indicated amount of time. Cells were directly plated on PYE plates containing kanamycin (25 μg/mL) and either chloramphenicol (1 μg/mL), tetracycline (2 μg/mL), or both to select for transferants.

To assay the benefit of GTA transfer during a single double-strand break, recipient cells bearing the inducible I-SceI double-strand break system and an empty pBXMCS-2 $kan^R$-marked plasmid and donor cells bearing either the $P_{xyl}$-*gafYZ* or $P_{xyl}$-empty plasmid were grown up to stationary phase in the presence of 0.2% glucose to repress the xylose promoter and kanamycin to maintain plasmid selection. Cultures were mixed 1:1 in 250 mL flasks and induced with 0.3% xylose and 500 μM vanillate. At indicated time points, cells were enumerated for viability and loss of vanillate sensitivity on plates containing chloramphenicol to select for recipient cells with or without 500 μM vanillate to select for vanillate-resistant cells. Cells were subsequently picked, genomic DNA was extracted, and the region containing the I-SceI cut site was sequenced by PCR and Sanger sequencing using primers oKRG523 and oKRG524.

### Bioinformatic analyses of sequence alignments and gene co-occurrences

Sequence alignment of *gafY* and *gafZ* from *C. crescentus* to *gafA* in *Rhodobacter capsulatus* were performed using MUSCLE [62]. *gafY* and *gafZ* from *C. crescentus* NC011916.1 were concatenated into 1 gene and aligned against *gafA* (rcc_01865) from *R. capsulatus* NC014034. Sequences were visualized using Jalview [63]. Co-occurrence of *gafY*, *gafZ*, *rogA*, the GTA capsid (*CCNA_02872*), the GTA terminase (*CCNA_02880*), and a putative GTA tail fiber (*CCNA_02456*) across 1,336 representative bacterial taxa using String-DB [64].

### Stationary phase benefit

Single colonies of the indicated strain were inoculated into 25 mL PYE in 250 mL flasks and grown at 30°C, shaking at 210 rpm. Cell survival was enumerated with 10-fold serial dilutions onto PYE plates. The marked $\Delta rogA::tet^R$, as opposed to a clean deletion, was used throughout for ease of quantifying donor and recipient cells where applicable. For co-incubation experiments, 25 mL of each indicated culture were grown overnight to stationary phase. Approximately 12.5 mL of the indicated cultures were mixed into fresh 250 mL flasks and cell survival was enumerated by serial dilutions on plates containing appropriate antibiotics. For experiments with filtered supernatant, stationary phase cultures of indicated strains were pelleted cells with centrifugation for 5 min at 8,000 rpm at 25°C and the supernatant was passed through a 0.20 μm SFCA filter (Corning) and used to resuspend pelleted recipient cells where indicated. Quantification of cells during co-culture with the *rec-* strain was performed with marked donors, either WT (*hfaB::kan^R*) or $\Delta rogA::tet^R$ cells, and markerless recipients, either WT or *rec-* strains. The co-incubation cultures were plated out to single colony and replica plated onto plates containing no antibiotic or the corresponding antibiotic of the donor and cells sensitive to antibiotic were enumerated.

### DNA damage treatment and disk assays

UV treatment was performed with a UV Stratalinker 2400. Treatment with "high" UV damage was equal to 5 J/m$^2$ (Fig 5F), while the *rec-* strain was exposed to 1 J/m$^2$ (Fig 5F). Exposure of cells in exponential phase was 1 J/m$^2$ (S6C Fig). For quantification of the *rec-* survival to UV when incubated with other strains, the *rec-* strain was enumerated using replica plating. Co-incubations were diluted out to single colonies before and after treatment on plates containing no antibiotic. Plates were then replica plated on plates containing antibiotic corresponding to the donor or plates without antibiotic, and colonies that only grew on antibiotic-free plates were enumerated, as these corresponded to the *rec-* strain. This was also performed with a markerless wild-type strain for comparison.

Soft agar chemical sensitivity assays were done as described previously [65] with some modifications. Each strain was grown to deep stationary phase in PYE (48 h of growth). A total of 60 μL of cells were added to 3 mL of warm PYE with 0.3% agar and poured onto a PYE agar plate. A 6-mm sterile filter paper disk was impregnated with 20 μL of the tested compound, air dried, and placed on top of the solidified soft agar. Zones of inhibition were measured after 48 h of incubation at 30°C. The distance between the disk and the edge of bacterial lawn growth was quantified using Fiji [66].

### Protein overexpression and purification

**His$_6$-MBP-GafY.**  Plasmid pML333::*his$_6$-mbp-gafY* was introduced into *E. coli* Rosetta (DE3) competent cells (Merck) by heat-shock transformation. A 40-mL overnight culture was used to inoculate 4 L of LB medium + carbenicillin + chloramphenicol. Cells were grown at

37˚C with shaking at 210 rpm to an $OD_{600}$ of approximately 0.4. The culture was then left to cool down to 28˚C before isopropyl-β-D-thiogalactopyranoside (IPTG) was added to a final concentration of 1.0 mM. The culture was left shaking for an additional 3 h at 28˚C before cells were harvested by centrifugation. Pelleted cells were resuspended in a buffer containing 100 mM Tris-HCl (pH 8.0), 300 mM NaCl, 10 mM imidazole, 5% (v/v) glycerol, 1 μL of Benzonase nuclease (Merck), 1 mg of lysozyme (Merck), and an EDTA-free protease inhibitor tablet (Merck). The resuspended cells were then lysed by sonication (10 cycles of 15 s with 10 s resting on ice in between each cycle). The cell debris was removed by centrifugation at 28,000 g for 30 min and the supernatant was filtered through a 0.45 μm filter disk. The lysate was then loaded into a 1-mL HisTrap column (GE Healthcare) that had been pre-equilibrated with buffer A [100 mM Tris-HCl (pH 8.0), 300 mM NaCl, 10 mM imidazole, and 5% glycerol]. Protein was eluted from the column using an increasing imidazole gradient (10 mM to 500 mM) in the same buffer. Fractions containing $His_6$-MBP-GafY were pooled and diluted to a conductivity of 16 mS/cm before being loaded onto a Heparin HP column (GE Healthcare) that had been pre-equilibrated with 100 mM Tris-HCl pH 8.0, 25 mM NaCl, and 5% glycerol. Protein was eluted from the Heparin column using an increasing salt gradient (25 mM to 1 M NaCl) in the same buffer. Fractions containing $His_6$-MBP-GafY were pooled and analyzed for purity by SDS-PAGE. Glycerol was then added to $His_6$-MBP-GafY to the final volume of 10%, and the protein was stored at -80 ˚C. Two mg of this protein was used to raise polyclonal antibody in rabbit (Cambridge Research Biochemicals, United Kingdom).

**$RogA$-$His_6$.** Plasmid pET21b::$rogA$-$his_6$ was introduced into *E. coli* Rosetta (DE3) competent cells (Merck) by heat-shock transformation. A 40-mL overnight culture was used to inoculate 4 L LB medium + carbenicillin + chloramphenicol. Cells were grown at 37˚C with shaking at 210 rpm to an $OD_{600}$ of ~0.4. The culture was then left to cool down to 28˚C before IPTG was added to a final concentration of 1.0 mM. The culture was left shaking for an additional 20 hours at 20˚C before cells were harvested by centrifugation. Pelleted cells were resuspended in a buffer containing 100 mM Tris-HCl pH 8.0, 300 mM NaCl, 10 mM imidazole, 5% (v/v) glycerol, 1 μL of Benzonase nuclease (Merck), 1 mg of lysozyme (Merck), and an EDTA-free protease inhibitor tablet (Merck). The resuspended cells were then lyzed by sonication (10 cycles of 15 s with 10 s resting on ice in between each cycle). The cell debris was removed by centrifugation at 28,000 g for 30 min and the supernatant was filtered through a 0.45 μm filter disk. The lysate was then loaded into a 1-mL HisTrap column (GE Healthcare) that had been pre-equilibrated with buffer A [100 mM Tris-HCl pH 8.0, 300 mM NaCl, 10 mM imidazole, and 5% glycerol]. Protein was eluted from the column using an increasing imidazole gradient (10 mM to 500 mM) in the same buffer. $RogA$-$His_6$ containing fractions were pooled and diluted to a conductivity of 16 mS/cm before being loaded onto a Heparin HP column (GE Healthcare) that had been pre-equilibrated with 100 mM Tris-HCl (pH 8.0), 25 mM NaCl, and 5% glycerol. Protein was eluted from the Heparin column using an increasing salt concentration (25 mM to 1 M NaCl) in the same buffer. $RogA$-$His_6$ fractions were pooled together and analyzed for purity by SDS-PAGE. Glycerol was then added to $RogA$-$His_6$ fractions to a final volume of 10%, and the protein was stored at −80 ˚C. Two mg of this protein was used to raise polyclonal antibody in rabbit (Cambridge Research Biochemicals, UK).

**$His_6$-GafZ and tagless GafY co-complex.** Plasmid pCOLADuet-1:$his_6$-$gafZ$ $gafY$ was introduced into *E. coli* Rosetta (DE3) competent cells (Merck) by heat-shock transformation. A 40-mL overnight culture was used to inoculate 4 L LB medium + kanamycin + chloramphenicol. Cells were grown at 37˚C with shaking at 210 rpm to an $OD_{600}$ of approximately 0.4. The culture was then left to cool down to 28˚C before IPTG was added to a final concentration of 1.0 mM. The culture was left shaking for an additional 3 h at 28˚C before cells were harvested by centrifugation. Pelleted cells were resuspended in a buffer containing 100 mM Tris-HCl

(pH 8.0), 300 mM NaCl, 10 mM imidazole, 5% (v/v) glycerol, 1 μL of Benzonase nuclease (Merck), 1 mg of lysozyme (Merck), and an EDTA-free protease inhibitor tablet (Merck). The pelleted cells were then lyzed by sonification (10 cycles of 15 s with 10 s resting on ice in between each cycle). The cell debris was removed by centrifugation at 28,000 g for 30 min and the supernatant was filtered through a 0.45 μm filter disk. The lysate was then loaded into a 1-mL HisTrap column (GE Healthcare) that had been pre-equilibrated with buffer A [100 mM Tris-HCl (pH 8.0), 300 mM NaCl, 10 mM imidazole, and 5% glycerol]. Protein was eluted from the column using an increasing imidazole gradient (10 mM to 500 mM) in the same buffer. Fractions containing the GafYZ complex fractions were checked by SDS-PAGE for purity. Proteins were subsequently used for SPR assay.

## Chromatin immunoprecipitation with deep sequencing (ChIP-seq)

*C. crescentus* cells were grown in PYE at 28 ˚C with shaking at 220 rpm in biological duplicate. When the cultures (60 mL) reached either exponential phase (for IP of RogA) or stationary phase (for IP of GafY or FLAG-GafZ), formaldehyde was added to a final concentration of 1%. Cells were further incubated at room temperature for 30 min, then the fixation was quenched using 0.125 M glycine for 15 min at room temperature. Cells were washed 3 times with 1× PBS (pH 7.4) and were resuspended in 1.5 mL of buffer 1 [20 mM K-HEPES (pH 7.9), 50 mM KCl, 10% glycerol, and EDTA-free protease inhibitors]. Subsequently, the cell suspension was sonicated on ice using a probe-type sonicator (8 cycles of 15 s with 15 s resting on ice, amplitude setting 8) to lyse cells and to shear the chromatin to below 1 kb. The cell debris was cleared by centrifugation (20 min at 13,000 rpm at 4˚C). The supernatant was then transferred to a new 2-mL tube and the buffer conditions were adjusted to 10 mM Tris-HCl (pH 8.0), 150 mM NaCl, and 0.1% NP-40. To immunoprecipitate FLAG-tagged GafZ, 100 μL of α-FLAG M2 agarose beads (Merck) was washed off their storage buffer by repeated centrifugation and resuspension in IPP150 buffer [10 mM Tris-HCl (pH 8.0), 150 mM NaCl, and 0.1% NP-40], beads were then introduced to the cleared supernatant and incubated with rotation at 4˚C overnight.

Protein A beads (Merck) with α-RogA and α-GafY polyclonal antibodies were employed to immunoprecipitate RogA and GafY, respectively. To immunoprecipitate RogA and GafY, 60 μL of α-RogA/α-GafY polyclonal antibodies were added to the cleared supernatant first, the mixture was then incubated overnight before protein A beads were added and further incubated for another 4 h.

After the incubation, beads were washed 5 times at 4˚C for 2 min each with 1 mL of IPP150 buffer, then twice at 4˚C for 2 min each in 1× TE buffer [10 mM Tris-HCl (pH 8.0) and 1 mM EDTA]. Immunoprecipitated protein–DNA complexes were then eluted twice from the beads by incubating the beads first with 150 μL of the elution buffer [50 mM Tris-HCl (pH 8.0), 10 mM EDTA, and 1% SDS] at 65˚C for 15 min, then with 100 μL of 1× TE buffer + 1% SDS for another 15 min at 65˚C. The supernatant (i.e., the ChIP fraction) was then aspirated from the beads and was incubated at 65˚C overnight to completely reverse the crosslinks. DNA from the ChIP fraction were then purified using a PCR purification kit (Qiagen) according to the manufacturer's instruction and eluted out using 50 μL of EB buffer (Qiagen). Subsequently, the purified DNA was used to construct barcoded libraries suitable for Illumina sequencing using the NEXT Ultra II library preparation kit (NEB). Barcoded libraries were pooled together sequenced on the Illumina Hiseq 2500 at the Tufts university genomics facility.

The generation and analysis of ChIP-seq profiles have been described previously [67]. Briefly, Hiseq 2500 Illumina short reads (50 bp) were mapped back to the *C. crescentus* NA1000 reference genome (NCBI Reference Sequence: NC011916.1) using Bowtie 1 [68]. Subsequently, the sequencing coverage at each nucleotide position was computed using BEDTools [69].

Finally, ChIP-seq profiles were plotted with the x-axis representing genomic positions and the y-axis is the number of reads per kilobase pair per million mapped reads (RPKM) using custom R scripts. ChIP-seq data are available at GEO accession number GSE184477.

## Co-immunoprecipitation (co-IP) of GafY and FLAG-tagged GafZ and immunoblot analysis

*C. crescentus* cells (80 mL culture) were grown at 28 ˚C to stationary phase before cells were harvested by centrifugation. Cell pellets were washed with 1× PBS (pH 7.4), resuspended in 1.5 mL of lysis buffer [50 mM Tris-HCl (pH8), 150 mM NaCl, 1% Triton X-100, EDTA-free protease inhibitors, 10 mg/mL lysozyme, and 1 μL of Benzonase], and incubated at 37 ˚C for 20 min. Subsequently, the cell suspension was sonicated on ice using a probe-type sonicator (6 cycles of 15 s with 15 s resting on ice, amplitude setting 8). The lysate was cleared from the cell debris by centrifugation (13,000 rpm for 20 min at 4 ˚C). A total of 50 μL of this supernatant (i.e., the INPUT fraction) was kept for immunoblot analysis. The remaining supernatant was mixed with 50 μL α-FLAG magnetic bead as instructed by the μMACS Epitope Tag Protein Isolation Kit (Miltenyi Biotec). From here, all the subsequent steps were performed according to the instruction from the μMACS kit, except for the washing step. The magnetic columns were washed instead with 2 mL of IPP200 buffer [10 mM Tris-HCl (pH 8.0), 200 mM NaCl, and 0.1% NP-40], followed by 0.5 mL of 1× TE buffer [10 mM Tris-HCl (pH 8.0) and 1 mM EDTA]. The immunoprecipitated proteins (i.e., the IP fraction) were eluted using 50 μL of elution buffer [50 mM Tris-HCl (pH 6.8), 50 mM DTT, 1% SDS, and 1 mM EDTA].

For immunoblot analysis, 10 μg of the INPUT fraction and 5 μL of the IP fraction were loaded on a 4% to 20% Novex WedgeWell SDS-PAGE gels (Thermo Fisher Scientific). Resolved proteins were transferred to polyvinylidene fluoride (PVDF) membranes using the Trans-Blot Turbo Transfer System (BioRad), and the membrane was incubated with a 1:2,500 dilution of an α-FLAG antibody (Merck), or a 1:5,000 dilution of an α-ParB antibody (custom antibody, Cambridge Research Biochemicals, UK), or a 1:300 dilution of an α-GafY antibody (custom antibody, Cambridge Research Biochemicals, UK). Subsequently, the membranes were washed twice in a 1× TBS + 0.005% Tween-20 buffer before being incubated in a 1:10,000 dilution of an HRP-conjugated secondary antibody. Blots were imaged using an Amersham Imager 600 (GE Healthcare).

## Surface plasmon resonance (SPR)

**Quantification of RogA-DNA interaction.**   Overlapping single-stranded oligomers that span the promoter region of *gafY* were dissolved to 100 μM in water (*gafY*_1–11_F-R). They were annealed together with their complementary oligos in an annealing buffer [10 mM Tris-HCl (pH 8.0), 50 mM NaCl, and 1 mM EDTA] to form double-stranded DNA before being diluted to a working concentration of 1 μM in HPS-EP buffer [0.01 M HEPES (pH 7.4), 150 mM NaCl, 3 mM EDTA, and 0.005% surfactant P20] for SPR experiments. SPR measurements were recorded at 25˚C using a Biacore 8K system (Cytiva). All experiments were performed using the Re-usable DNA Capture Technique (ReDCaT) as described previously [70]. Briefly, ReDCAT uses a sensor chip SA Cytiva that has streptavidin pre-immobilized to a carboxymethylated dextran matrix, to which a 20-base biotinylated ReDCaT linker is immobilized. This chip is then used to immobilize $P_{gafY}$- dsDNA on the chip surface as each dsDNA also contain a single-stranded overhang complimentary to the ReDCaT linker on the surface. The DNA to be tested was flowed over the test flow cell on the chip at a flow rate of 10 μL/min and it annealed through the complementary DNA to the ReDCaT linker, thus was immobilized on the surface of the chip.

Purified RogA-His$_6$ protein, pre-diluted in HBS-EP buffer, was then flowed over the chip surface (both the surface with the immobilized DNA and a blank surface as a control), followed by HBS-EP buffer to observe RogA-His$_6$ dissociation from the DNA. The chip was then regenerated using 1 M NaCl and 50 mM NaOH to remove any residual RogA-His$_6$ protein and the test DNA. The cycle can then be repeated as many times as desired with new test DNA and protein concentration. The Biacore 8K has 8 channels and 2 flow cells so for each cycle, 8 different DNA samples can be tested (and each referenced against a reference flow cell). The SPR signal (response units) was monitored continuously throughout the process. All sensorgrams recorded during ReDCAT experiments were analyzed using Biacore Insight Evaluation software version 3.0.11.15423 (Cytiva). Data were then plotted using Microsoft Excel.

**Quantification of GafYZ-DNA interaction.** Overlapping single-stranded oligomers that spans the promoter region of *CCNA_02880* (GTA terminase) were dissolved to 100 μm in water (*02880*_1–9_F-R). They were annealed together with their complementary oligos in an annealing buffer [10 mM Tris-HCl (pH 8.0), 50 mM NaCl, and 1 mM EDTA] to form double-stranded DNA before being diluted to a working concentration of 1 μM in HPS-EP buffer. A similar ReD-CaT procedure was used to measure the interaction between the GafYZ complex and immobilized DNA, except that a custom binding buffer TSE-T buffer [10 mM Tris-HCl (pH 8.0), 200 mM NaCl, 1 mM EDTA, and 0.005% Tween 20] instead of HPS-EP buffer was used instead.

## Purification of GTA particles and analysis by transmission electron microscopy (TEM)

*C. crescentus ΔrogA* cells (in 200-mL culture) were grown in PYE at 28 ˚C to stationary phase. Cells were pelleted by centrifugation (8,000 rpm for 10 min at 4 ˚C). The supernatant that contained GTA particles was then transferred to a fresh 500-mL bottle and was filtered twice using a 0.22-μm filter funnel (Sartorius). The supernatant was concentrated using a 100-kDa MWCO Amicon concentrator (Merck) to approximately 25 mL. GTA particles in this concentrated supernatant were precipitated by incubating with a 5xPEG/NaCl solution [20% PEG-8000 and 2.5 M NaCl] on ice for 30 min. Precipitated GTA particles were collected by centrifugation (8,000 rpm for 20 min at 4˚C) and were resuspended in 200 μL storage buffer [50 mM Tris pH 8.0, 150 mM NaCl, and 5% glycerol].

For TEM analysis, 3 μL of the purified GTA particles was pipetted on a 400-mesh copper grid (EM Resolutions) that had been glow discharged for 20 s at 10 mA in an Ace 200 (Leica Microsystems). After 60 s, excess solution was wicked away using Whatman No. 1 filter paper and leftover samples on the grid were stained using 2% (w/v) uranyl acetate solution. Grids were imaged using a Talos F200C transmission electron microscope (Thermo Fisher Scientific) operated at 200 kV, equipped with a 4 k OneView CMOS detector (Gatan, UK).

## Supporting information

**S1 Fig. Activation of the *didA* promoter by DNA damage.** β-galactosidase assay measuring transcriptional activity of a *didA* reporter in WT, Δ*rogA* and Δ*driD* cells grown up to early stationary phase and treated with or without 15 μg/mL zeocin for 45 min (*n* = 3, error bars indicate SD). * = *p*-value <0.05 of indicated comparisons. ns = not significant. Data are available in S1 Data.
(PDF)

**S2 Fig. Conservation of the *C. crescentus* GTA locus.** (A) Comparison of the main GTA clusters from *Caulobacter crescentus* NA1000 and *Rhodobacter capsulatus* SB1003. The corresponding homolog of each *C. crescentus* gene in *R. capsulatus* is shown with the same color.

(B) Amino acid sequence alignment of GafY and GafZ from *C. crescentus* to GafA in *R. capsulatus*. Identical amino acids are shaded black, with similar amino acids shaded in gray. (C) Co-occurrence of GTA cluster genes (all genes >100 AA), *gafY*, *gafZ*, *rogA*, and the GTA tail fiber homolog found across the α-proteobacteria. Co-occurrence is shown as the (# of genomes with at least 1 homolog of both gene X and gene Y)/(# of genomes with a homolog of gene Y). Non-symmetry across the diagonal plane results from the change in # of genomes in the denominator. Data are available in S1 Data. (D) β-galactosidase assay measuring transcriptional activity of a *didA* reporter ($P_{didA}$-*lacZ* and Δ*didA*) bearing a high-copy plasmid with the xylose promoter driving expression of either nothing (empty), *gafY* alone, or both *gafY* and *gafZ* in their native arrangement. Cells were grown up to stationary phase ($OD_{600}$ = ~1.3) and induced with or without xylose for 3 h. (*n* = 3, error bars indicate SD). * = *p*-value <0.05 of indicated comparisons. Data are available in S1 Data.
(PDF)

**S3 Fig. GafY, GafZ, and RogA ChIP-seq profiles.** (A) Individual ChIP-seq profiles of GafY with an anti-GafY antibody in either Δ*rogA* (black) or Δ*rogA* Δ*gafY* (gray). Profiles were plotted with the x-axis representing genomic positions and the y-axis representing the number of reads per kilobase pair per million mapped reads (RPKM) using custom R scripts. Corresponding gene annotations are shown with comparative RNA-seq data as shown before. Data are available in S1 Data. (B) ChIP-seq profile of the 1 identified peak of FLAG-GafZ with an anti-FLAG antibody in either Δ*rogA* *gafZ*::*FLAG-gafZ* (dark blue) or Δ*rogA* (light blue). Profiles were plotted as in S3A Fig. Data are available in S1 Data. (C) Individual ChIP-seq profiles of the 2 identified peaks of RogA with an anti-RogA antibody in either WT (purple) or Δ*rogA* (light purple). Profiles were plotted as in S3A Fig. Data are available in S1 Data.
(PDF)

**S4 Fig. DNA binding by RogA and GafY•GafZ.** (A) Surface plasmon resonance analysis of purified RogA binding to several overlapping segments of the *gafYZ* promoter. Data are available in S1 Data. (B) SPR response of RogA-His$_6$ to the 3 best binding probes as a function of protein concentration. Corresponding $K_d$ is shown. Data are available in S1 Data. (C) Probe position relative to the *gafYZ* promoter with potential inverted repeats highlighted in yellow. (D) SPR response of purified GafY•GafZ complex to various DNA probes corresponding to the GTA cluster promoter. Probe position relative to the promoter is shown above. Data are available in S1 Data.
(PDF)

**S5 Fig. Analysis of GTA-packaged DNA.** (A) SalI restriction digestion of purified GTA DNA run on a 6% TBE gel. (B) Quantitative PCR of different genomic regions in the GTA DNA. Relative enrichment of each locus is normalized to quantification using genomic DNA as input. Data are available in S1 Data. (C) Time course of quantification of colony-forming units of WT and Δ*rogA* cells starting at OD 0.75, with time points every 40 min. * = *p*-value <0.05 comparing data of same time point. Data are available in S1 Data. (D) Propidium iodide (1 μm final concentration) staining of cells overexpressing *gafYZ* from a high-copy vector ($P_{xyl}$-*gafYZ*) with phase images. Scale bar = 10 μm. (E) Approximately 5% to 45% sucrose gradients of purified GTA samples from cells with a plasmid bearing either $P_{xyl}$-empty or $P_{xyl}$-*gafYZ*. (F) SDS-PAGE gel of proteins extracted from GTAs purified with ultracentrifugation through sucrose gradients. (G) Transmission electron micrographs of PEG-precipitated filtered supernatant from cells overexpressing *gafYZ* from the $P_{xyl}$-*gafYZ* high-copy plasmid in stationary phase. Scale bar = 100 nm. Particles noted with white arrows.
(PDF)

**S6 Fig. GTA-mediated survival following DNA damage.** (A) Survival of the wild-type (*hfaB*::*kan*<sup>R</sup>) strain in combination with WT, Δ*rogA*, Δ*rogA* Δ*driD*, or Δ*driD* cells at 48 h and 96 h in stationary phase. (B) Survival after 96 h of representative replicates of either an isogenic WT strain or of surviving WT cells that had previously co-incubated with Δ*rogA* cells for 96 h. (C) Survival of exponential WT, Δ*rogA*, or Δ*rogA* Δ*gta* cells to UV exposure (1 J/m$^2$).
(PDF)

**S1 Table. Plasmids.**
(PDF)

**S2 Table. Strains.**
(PDF)

**S3 Table. DNA oligonucleotides.**
(PDF)

**S1 Data. All numerical data underlying graphs and plots shown in main and supporting figures.**
(XLSX)

**S1 Raw Images. Uncropped images for gels in Fig 2.**
(PDF)

**S2 Raw Images. Uncropped images for gels in S5 Fig.**
(PDF)

# Acknowledgments

We thank Idan Frumkin, Chantal Guegler, Mathilde Guzzo, and Rosemary Redfield for helpful discussions and comments on the manuscript. We also thank Clare Stevenson and Jake Richardson for help with SPR and TEM analysis.

# Author Contributions

**Conceptualization:** Kevin Gozzi, Ngat T. Tran, Joshua W. Modell, Tung B. K. Le, Michael T. Laub.

**Data curation:** Kevin Gozzi, Ngat T. Tran, Michael T. Laub.

**Formal analysis:** Kevin Gozzi, Ngat T. Tran, Michael T. Laub.

**Funding acquisition:** Tung B. K. Le, Michael T. Laub.

**Investigation:** Kevin Gozzi, Ngat T. Tran, Joshua W. Modell, Tung B. K. Le.

**Methodology:** Kevin Gozzi, Ngat T. Tran.

**Project administration:** Tung B. K. Le, Michael T. Laub.

**Resources:** Kevin Gozzi, Tung B. K. Le.

**Supervision:** Tung B. K. Le, Michael T. Laub.

**Validation:** Kevin Gozzi.

**Writing – original draft:** Kevin Gozzi, Ngat T. Tran, Michael T. Laub.

**Writing – review & editing:** Kevin Gozzi, Tung B. K. Le, Michael T. Laub.

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
