## [Editor Report · Decision Letter 0]

26 Jan 2022

Dear Michael, 

Thank you for submitting your manuscript entitled "Gene transfer agents promote survival and DNA repair during stationary phase for Caulobacter crescentus" for consideration as a Research Article by PLOS Biology.

Your manuscript has now been evaluated by the PLOS Biology editorial staff and I am writing to let you know that we would like to send your submission out for external peer review.

Once your full submission is complete, your paper will undergo a series of checks in preparation for peer review. Once your manuscript has passed the checks it will be sent out for review. To provide the metadata for your submission, please Login to Editorial Manager (https://www.editorialmanager.com/pbiology) within two working days, i.e. by Jan 28 2022 11:59PM.

If your manuscript has been previously reviewed at another journal, PLOS Biology is willing to work with those reviews in order to avoid re-starting the process. Submission of the previous reviews is entirely optional and our ability to use them effectively will depend on the willingness of the previous journal to confirm the content of the reports and share the reviewer identities. Please note that we reserve the right to invite additional reviewers if we consider that additional/independent reviewers are needed, although we aim to avoid this as far as possible. In our experience, working with previous reviews does save time. 

If you would like to send previous reviewer reports to us, please email me at dummarino@plos.org to let me know, including the name of the previous journal and the manuscript ID the study was given, as well as attaching a point-by-point response to reviewers that details how you have or plan to address the reviewers' concerns. 

Given the disruptions resulting from the ongoing COVID-19 pandemic, please expect some delays in the editorial process. We apologise in advance for any inconvenience caused and will do our best to minimize impact as far as possible.

Kind regards,

Dario

Dario Ummarino, PhD

Senior Editor

PLOS Biology

dummarino@plos.org

---

## [Decision Letter · Decision Letter 1]

22 Mar 2022

Dear Michael,

Thank you for submitting your manuscript "Gene transfer agents promote survival and DNA repair during stationary phase for Caulobacter crescentus" for consideration as a Research Article at PLOS Biology. Your manuscript has been evaluated by the PLOS Biology editors, an Academic Editor with relevant expertise, and by several independent reviewers.

As you will see in the reviews attached below, all reviewers recognize the importance of your study. However, they also raise concerns about the experimental design and the empirical support of your conclusions. Having discussed the reviews with the Academic Editor, we would like to point out that we will be expecting more statistical rigor and more rigorous tests for the hypotheses proposed for the role of gene transfer agents, since that represents a central aspect of the novelty of your findings. 

In light of the reviews, we would welcome re-submission of a much-revised version that takes into account the reviewers' comments. We cannot make any decision about publication until we have seen the revised manuscript and your response to the reviewers' comments. Your revised manuscript is also likely to be sent for further evaluation by the reviewers.

We expect to receive your revised manuscript within 3 months. 

**IMPORTANT - SUBMITTING YOUR REVISION**

*Re-submission Checklist*

*Published Peer Review*

*PLOS Data Policy*

*Blot and Gel Data Policy*

Sincerely,

Dario

Dario Ummarino, PhD

Senior Editor

PLOS Biology

dummarino@plos.org

REVIEWS:

Reviewer #1: This manuscript describes a novel gene transfer agent (GTA) in Caulobacter, its characterization, and the test of its role. GTAs are intriguing elements, probably derived from phages, that package the DNA of dying cells in viral-like particles. These particles can inject the DNA in other cells. The role of these elements is yet unclear and only a handful of model systems are known. Here, a novel one is introduced. This novel system has many similarities with known systems: its locus is smaller than prophages, it is regulated by systems homologous to known ones, it packages small DNA fragments (~8 kb) more or less randomly, triggers cell lysis, and can transfer the DNA to cells of the same species. The first 5 pages of results describe these results and are an interesting and solid contribution to the specific domain of GTA research. The final results concerns the test of hypothesis for the role of GTAs and are of broader interest. My review focus on the latter.

The authors consider 3 hypothesis. (1) GTAs allow cells to acquire and spread new beneficial alleles, (2) their production releases beneficial nutrients (3) GTAs allow cells to acquire and use wild-type DNA to repair or revert deleterious mutations. 

1) The test of the first hypothesis is not convincing. By co-incubating strains and their DrogA mutants the authors test if this allows the former to extend survival in the stationary phase. The details presented in results are so succinct that I'm uncertain of where this is described in methods. There seems to be no evidence that in this setup there are adaptive mutations to be transferred. It is also unclear why testing only in stationary phase (elements are produced in these conditions but can disperse and infect in other conditions). It's not clear to me if one can conclude anything from the experiments. If this hypothesis is to be tested, there should be clear knowledge of an advantageous mutation to be transferred between cells. And there should be tests that this can or cannot take place under a range of experimental conditions.

2) It's unclear why GTA would be a strategy to release nutrients. Viral particles do not enter the cell and they are very costly for the producer. Just the delivery of DNA does not seem an interesting strategy to feed other bacteria. The hypothesis seems not to work, so this is a minor detail. 

3)The authors conclude that their results "suggest that GTAs can improve a recipient cell's ability to survive DNA damage ". I agree with the conclusion that they can. However, I was surprised that they failed to mention similar tests of the broader hypothesis that uptake of conspecific DNA favors DNA repair. This is one of the most studied hypotheses for the existence of natural transformation and has been reviewed several times before (for example Claverys, Annual Rev Microbiol, 06). There are also several studies showing that DNA damaging agents promote transformation hinting at a connection between acquisition of DNA and DNA repair. The results presented here should be integrated in the context of this literature. It should also be added that the hypothesis DNA for repair has been criticized on several other grounds, notably the likelihood that the incoming DNA matches the region in need of repair, which seems low since the genome is several Mb and the GTA carries less than 10 kb. Maybe the present results could add interesting facts to the existing debate, but at this stage that's unclear.

-There is a theoretical complication specific to the GTAs. In transformation, bacteria program a response to acquire DNA eventually useful for repair. But in GTAs, the choice is not on the recipient, but on a bacterium that dies and offers its DNA. From the point of view of evolutionary biology this is much harder to explain for hypothesis #3 than for hypothesis #1. These issues are briefly mentioned in the discussion, but there is no integration with past works, no reference to models or literature, and this complicates the identification of the novelties in these results (beyond the fact that it deals with GTAs and not with other mechanisms of transfer). 

-L69. "Modelling shows that it is more favourable for a population if cells that acquire a beneficial allele simply grow and propagate the allele via vertical inheritance rather than distribute the allele via GTAs. " This is given, if I understand correctly, as evidence that GTAs may not be adaptive for the transfer of novel functions. I find the argument very weak (if the cell is dying, saving some genes can be adaptive enough), but more importantly I don't see how the argument works better for the hypothesis #3 than for hypothesis #1. Why should bacteria carry elements whose death will allow the repair of DNA of potentially non-producer cells? If their genes are being reproduced (hypothesis #1) this seems easier to understand than if they are not (hypothesis #3).

There is a pervasive lack of statistical analysis of the experimental results. Conclusions about differences (significant or non-significant) should be supported by statistical analyses. 

-Figure 1 In C to test differences and in E for the threshold of fold change 4 (what does it meand in statistical terms).

-Figure 2E has no statistics and one is informed that "Note only a portion of the plot is shown". That's not acceptable. The full plot should be in supplementary material and statistics are needed.

-Figure 4E needs stats.

-Same for figure 5 and 6. 

-Figure S1C needs statistical analyses. At the very least one would need to know the denominator of the equation (the meaning of dark blue is very different if the latter is large or small). Statistical tests should also be corrected for phylogeny.

Minor comments. 

-L101. Unclear to me. If the regulator is LexA-like and its associated with response to DNA damage, why is it stated that it is SOS-independent? Maybe this is just a variant of the standard SOS response? 

Reviewer #2 (Andrew S. Lang): The manuscript by Gozzi and colleagues presents a broad range of molecular and microbiology experiments that demonstrate the bacterium C. crescentus produces a gene transfer agent (GTA). They also demonstrate that production of this GTA leads to increased cell survival in the stationary phase of lab culture growth and that the GTAs allow DNA repair via homologous recombination. This is particularly noteworthy because HR has long been speculated to be a potential benefit to cells from producing GTAs but has never been proven. The work has great scope and importance, and the experiments are well designed and sound. The manuscript is well written, clear and has a good, logical flow.

One key (and simple) experiment that is missing, which is a definitive experiment to confirm that transformation is not somehow involved here, is the demonstration that the gene transfer still happens after the culture supernatant or filtrate is treated with DNase to degrade any free DNA released into the environment.

Is there any indication that there might be translational coupling for gafY and Z? I.e., could the gafY knockout be polar on gafZ?

If I zoom way in on figure 2B (xylose) it looks like there might be a very faint GTA DNA band in the gafY lane?

I am a bit confused about some of the ChIP-seq data. It seems the implication is that GafZ is only involved in transcription of the one locus, whereas GafY is involved in transcription of all relevant loci. This strikes me as a bit off and could be investigated or discussed further. Presumably only GafY binds to the DNA whereas GafZ does not - is GafZ then captured by cross-linking to another protein that is bound to the DNA?

Another puzzle is the DNA transfer without tails. To my knowledge this is completely unprecedented for a "tailed phage" type of phage-like structure and there are tail genes expressed in the GTA cluster. Given this, it makes the DNase experiment mentioned previously even more relevant.

It would be nice if the authors could provide some speculation on how the cells lyse to release the GTAs.

It would also be nice to have some additional information on some of the additional loci that are upregulated with the GTA cluster - e.g., any clues based on homology what gene 1082 might encode?

A few minor points:

line 139 has something off with the wording

line 146 - at this point I feel like the DNA band is still putatively the GTA DNA?

line 408 - it is mentioned later, but since cell density is mentioned here it could be added about known quorum sensing effects on GTA production in Rhodobacter and Dinoroseobacter?

Reviewer #3: Gene transfer agents promote survival and DNA repair during stationary phase for Caulobacter crescentus by Gozzi et al.

 In this manuscript, the Laub lab identified a GTA cluster in the bacterium Caulobacter crescentus. The authors successfully described this novel GTA and found its regulation mechanism, via a novel GTA regulator, the transcription factor RogA, that represses GafY and GafZ, homologs of GafA in R. capsulatus. They also found that GTA particles production gives a selective advantage to survive when cells reach stationary phase and when they suffer DNA damage. This work is an important contribution to the field of GTA. It also provides important insights into DNA transfer and cell survival in Caulobacter.

 I have very few concerns about this manuscript. I think the data presented here are convincing, the experiments well-thought, and the conclusions solid. However, I still have a few suggestions that would improve the quality of the manuscript.

Major comments:

1. The authors suggest that " RogA indirectly regulates didA by somehow modulating DriD's expression or activity, possibly by causing an accumulation of double-stranded DNA breaks... " (lines 110-112) . This is an interesting point that deserves more attention. If RogA, DidA and DriD are in the same pathway:

a. Is RogA also regulated by DNA damage or is it upstream in the DriD/DidA regulation pathway? What is PdidA promoter activity in the presence of DNA damage in the strains tested in Fig 1C?

b. DidD and DidA inhibit cell division (Fig. 1A), so how is cell division in the ∆rogA mutant? And what about in gafYZ-overexpression strains? From Fig. 4C, it seems like the Pxyl-gafZY strain looks fine, but is this consistent with the described DriD/DidA connection? 

c. Finally, how do ∆rogA cells cope with DNA damage in the DriD/DidA context? The last result section ("GTAs promote survival following DNA damage") shows a substantial increase in survival of the ∆rogA strain after DNA damages, and it seems to be entirely GTA-dependant. How does it connect back to DriD/DidA?

2. Among the same lines, the authors clearly show here that a ∆rogA mutant and a gafYZ overexpressing strain have similar phenotypes, as RogA represses gafYZ expression. Is it a direct correlation between DidD/DidA, gafZY expression and GTA production? What is the PdidA promoter activity when gafY and gafZ are overexpressed?

3. It has been shown previously that, in R. capsulatus, GafA activity and GTA are regulated by CtrA. CtrA has been extensively studied in Caulobacter. Is this master regulator involved in GTA regulation in Caulobacter? In the discussionm, the authors state that "... the CckA-ChpT-CtrA phosphorelay [...] regulate(s) GTA synthesis in Rhodobacter (30-34), but not in Caulobacter." (lines 448-450), but they need to provide more context / data to back-up this claim for the non-Caulobacter expert readers. 

4. The authors showed here that Caulobacter GTA are mostly produced in stationary phase. What is the mechanism? In other species, GTA can be regulated by ppGpp and/or nutrient starvation. Is it the case in Caulobacter? All experiments performed in the present work were done using rich PYE medium. What happens to GTA production when Caulobacter cells are starved? What about in a ∆spoT strain that cannot produce ppGpp? 

5. Fig. 3B clearly shows that overexpressing gafY and gafZ increases cell death. The authors then provide quantification for "lysis" (measured as released protein concentration as a proxy) in Fig. 3C. The experiments are done in PYE, which is mainly made of peptone and yeast extract. How accurate measurement can be provided using this medium (the method section lacks information, see minor comments)? Furthermore, as overexpressing gafYZ rises the number of GTA produced, how can the authors conclude that the increase in extracellular protein content they detect is solely linked to lysis, and not GTA particle release in other way? It seems like other techniques to assay lysis would be more appropriate (such as enzymatic assays of cytoplasmic proteins in cell-free supernatants, western blot using a known cytoplasmic protein, or live/dead staining, for example).

6. What is the cell viability of a ∆rogA mutant over time compared to WT? Fig. 3D only provides extracellular protein content and cell density. Released protein content spikes in the ∆rogA mutant, while cell density is similar for both WT and ∆rogA. Can the author comment on that? What are the CFUs data for both strains under those conditions? 

7. Fig 4 A-B: It would be interesting to see the results for the SYBR-gold staining for the ∆rogA mutant and the TEM for the supernatant of the gafYZ overexpressing strain (especially as the authors used both strains as "GTA-donors" later on (see point #10)).

8. "No visible tails" (line 232) could be detected in the TEM images (Fig. 4B), while the genes encoding the tail proteins are present in the GTA encoding locus (Fig.1D-E). Can the author comment on that?

9. The authors nicely show GTA-mediated DNA transfer from one Caulobacter crescentus CB15N to another CB15N cell. What about DNA transfer to other Caulobacter strains and closely related Caulobacterales species? 

10. Fig. 5 and 6: Why are "GTA-producing donors" different in the all the co-cultures experiments. Is there a rationale to use ∆rogA strains in some experiments and gafYZ overexpressing strains in other? 

Minor comments:

1. The method section sometimes lacks key information (please read all and add relevant details when needed, important for the non-Caulobacter experts). For example:

a. The authors worked with a marked rogA mutant (∆rogA::tetR), while clean in-frame unmarked deletions are commonly used in C. crescentus. Please, briefly explain the rationale for not choosing to do a unmarked deletion.

b. I missed the method used for the quantification of released protein content as a proxy for cell lysis.

c. The authors chose to integrate PdidA-lacZ, Pxyl-gafYZ and other markers to the hfaB locus in the genome. Can the authors briefly explain their choice.

d. The method used to quantify each population in co-cultures (used to generate the data shown in Fig.5) is unclear and lacks details.

e. I missed the methods for UV damage induction (exposure time?).

2. Lines 138 - 139: " total DNA was isolated and separated by gel electrophoresis from cells in stationary phase, which was when we didA expression was highest in the ∆rogA mutant". Please rephrase.

3. Fig. 3E: It would be nice to see the total plot, with the relevant log2 from 2 to 15 portion as an insert.

4. Fig 4E: Please provide the number of replicates done in the legend and add error bars to the graphs (as it is done for all other experiments).

5. I am assuming that data illustrated in Fig. 5C and 5D are generated in a similar way (the manuscript is lacking information about this, see minor point 1d above). For a better readability, I would suggest for both experiments to be presented the same way. 

6. Same comment for Fig. 5E and 5F.

7. Fig. 6A: the chloramphenicol resistance cassette should be depicted in the scheme describing the experiment.

8. Fig. 6D: Is the difference in survival against EMS significant? Providing relevant statistics here might be a nice addition.

---

## [Decision Letter · Decision Letter 2]

25 Jul 2022

Dear Dr. Laub,

Thank you for your patience while we considered your revised manuscript "Gene transfer agents promote survival and DNA repair during stationary phase for Caulobacter crescentus" for publication as a Research Article at PLOS Biology. This revised version of your manuscript has been evaluated by the PLOS Biology editors, the Academic Editor, and the original reviewers.

Based on the reviews, we are likely to accept this manuscript for publication, we highly recommend you address the remaining points raised by the reviewers in order to make the manuscript more accessible to a wide audience. Please also make sure to address the following data and other policy-related requests.

1. DATA POLICY:

Regardless of the method selected, please ensure that you provide the individual numerical values that underlie the summary data displayed in the following figure panels as they are essential for readers to assess your analysis and to reproduce it: Figures 1CE, 2EFH, 3ACD, 4DEFG, 5ADE, 6D, and Supplementary Figures S1, S2CD, S3, S4ABD, S5BC.

**Please also ensure that figure legends in your manuscript include information on where the underlying data can be found, and ensure your supplemental data file/s has a legend.**

2. Please provide a blurb which (if accepted) will be included in our weekly and monthly Electronic Table of Contents, sent out to readers of PLOS Biology, and may be used to promote your article in social media. The blurb should be about 30-40 words long and is subject to editorial changes. It should, without exaggeration, entice people to read your manuscript. It should not be redundant with the title and should not contain acronyms or abbreviations.

3. We suggest a modification in the title to make it more accessible to a wide audience: "Prophage-like gene transfer agents promote Caulobacter crescentus survival and DNA repair during stationary phase"

We expect to receive your revised manuscript within two weeks.

*Published Peer Review History*

*Press*

Sincerely,

Paula

---

Senior Editor,

pjaureguionieva@plos.org,

PLOS Biology

Reviewer remarks:

Reviewer #2: Andrew S. Lang

Reviewer #1: I thank the authors for having tackled my comments and criticisms seriously. I appreciate the novel analyses, statistical tests and discussion. I've no further objections or criticisms.

As a note to the authors response (this is without impact to the evaluation of the paper): "Perhaps most notable is the fact that cells acquiring DNA by transformation cannot 'know' the origin of the incoming DNA and whether it is conspecific or not. " This statement is incorrect. In Proteobacteria several systems exist of uptake signal sequences allowing just that. In Firmicutes this recognition can be achieved by pheromones. So I do think there are extensive parallels between acquiring DNA by transformation and GTA. But I'm fine with the current level of the discussion.

Reviewer #2: The authors have thoroughly responded to the comments made during the initial reviews. I am only left unsatisfied with one point, which is related to the structure of the GTA particles and explanation given for the lack of tails. The authors state that it is possible there are tails there but that they are too short to observe in the TEM images. Is the implication that these GTAs are now akin to podophage structures? I.e., as opposed to siphophage like their relatives in other species (Rhodobacter and Dinoroseobacter)? The C. crescentus GTA cluster contains some of the tail gene homologs, but not all - is it proposed that this reduced gene set allows gene transfer, but in the absence of an extended tail structure? Considering the high resolution structure that is now available for RcGTA (Bardy et al 2020), how do we reconcile the potential tail protein functions present an absent and how DNA might be transferred? This is where information about which of these genes are required or not for the GTA activity to occur, but I do not want to suggest this as a condition for this manuscript - however, it seems to warrant more discussion at least and some mention of potential future steps to address this. It actually has some fairly broad implications if these particles can function for DNA transfer with such seemingly incomplete tails. I believe it is also the case that some TEM preparation/staining techniques can cause some phage particles to lose their tails so perhaps this should be investigated and discussed?

---

## [Editor Report · Decision Letter 3]

9 Aug 2022

Dear Dr. Laub,

Thank you for the submission of your revised Research Article "Prophage-like gene transfer agents promote Caulobacter crescentus survival and DNA repair during stationary phase" for publication in PLOS Biology. On behalf of my colleagues and the Academic Editor, Harmit Malik, I am pleased to say that we can in principle accept your manuscript for publication, provided you address any remaining formatting and reporting issues. These will be detailed in an email you should receive within 2-3 business days from our colleagues in the journal operations team; no action is required from you until then. Please note that we will not be able to formally accept your manuscript and schedule it for publication until you have completed any requested changes.

PRESS

Sincerely, 

Paula

---

Senior Editor

PLOS Biology
